



**Urban sewershed overflow analysis using super-resolution weather radar**
**rainfall**
**J. Y. Hyun[1], T. D. Rockaway[1] and M. N. French[1]**
[1]{University of Louisville, Louisville, Kentucky}
Correspondence to: J. Y. Hyun (j0hyun01@cardmail.louisville.edu)
**Abstract**
In urban areas, a prevalence of combined sewer systems (CSS) exist that carry both
storm water runoff and sanitary sewer flows in a single pipe, these system are considered
combined sewers.   In the absence of rainfall-runoff most of these systems function
adequately, however CSS capacity is typically inadequate to carry peak stormwater runoff
volume.   In order to minimize sewage flooding into streets and backups into homes and
businesses, most CSS (as well as separate sanitary sewer system) are designed to overflow
into surface waters such as streams and rivers, lakes and seas.
The primary factor causing overflows to occur is excessive precipitation with rainfall
at high intensity or high volume. A framework for the application of radar-rainfall data to
identify rainfall characteristics (spatial and temporal) associated with CSO events is presented
in this work. An innovative component of the work is the identification of a relationship
between weather radar reflectivity and ground-level rainfall. Additionally, the sewershed specific
radar-rainfall region is extracted for use in defining CSO triggering rain event characteristics at
sewershed spatial scale.   Results show underestimation of rainfall is more problematic than
overestimation. An optimal radar-rainfall relationship is developed to address reflectivity (Z) to
rainfall (R) transformation and improves rainfall estimates in the higher reflectivity range (greater





or equal to 46 dBZ). A large portion of quarter-hourly rainfall accumulations occurring at lower
radar reflectivity, less than 46 dBZ, indicate optimal reflectivity-rainfall (Z-R) relation
parameters range from [300, 1.4] to [250, 1.4] for convective storm types, and [250, 1.2] to
{[80, 1.4], [120, 1.4]} for tropical rainfall types.  Both rainfall and overflow events are
identified using criteria proposed by United States Environmental Protection Agency
(USEPA) to define the physical continuity of natural rainfall processes and the corresponding
hydrologic response. The methodology framework is illustrated using an urban sewershed,
denoted as CSO 130, located in Louisville, Kentucky (USA).  The role of specific rainfall
event characteristics: total volume, intensity, duration, continuity, and storm types are shown
to govern the overflow in the approximately 13-ha (30-ac) sewershed.  Through discriminant
analysis, the coupled rainfall and overflow events are categorized by overflow severity.
Results indicate that use of fine-resolution radar-rainfall in this urban hydrologic system can
reveal insights for planning CSO control and prevention measures for specific rainfall event
regimes.
**1. Introduction**

In many urban areas, combined sewer systems (CSS) carry both storm water runoff

and sanitary sewer flows in a single pipe.  In the absence of rainfall-runoff most CSS
adequately convey waste water flow, however system capacity may be overwhelmed when it
must also transport significant stormwater runoff.  In order to prevent sewage from flooding
streets and backups into homes and businesses, most CSS (as well as separated sanitary sewer
systems) are designed to overflow into surface waters such as streams and rivers, lakes and
seas.  This overflow occurrence is considered a combined sewer overflow (CSO) event and
has a detrimental impact on aquatic environments and degrades downstream water quality.



In the United States (USA), regulations were established to eliminate CSO events in

urban areas (EPA 1994).  Although CSS are now considered an outdated approach to waste
water collection, these legacy water collection systems form a considerable portion of the
sanitary sewer network in the United States.  It is estimated that 860 communities across the
USA are served by combined sewer systems with over 10,000 CSO outfalls directed into
natural surface waters.  These communities include approximately 40 million people in more
than 30 states (EPA 2004).  The direct solution to eliminate overflow occurrence through
modification or replacement of CSS with separate sewer and storm drains is cost prohibitive,
disruptive to the community, and difficult or infeasible to accomplish in existing urban
environments (Lyandres and Welch, 2012). When a CSO event occurs, the effect on receiving
waters can be significant.  The overflow transports microbial pathogens, oxygen depleting
substances, suspended solids, toxics, nutrients, and debris including floatables and trash
directly into the natural aquatic environment (EPA 2004).  Furthermore, in most urban areas
CSO occurrence is often a sudden phenomenon, due to both characteristics of the triggering
rain storm and hydrologic conditions in the sewershed, resulting in a surge of runoff (Romnée
et al. 2015).

Understanding the role of CSO triggering precipitation events is considered in

previous studies on the relationship between rainfall and CSO occurrence.  Recent work by
Mailhot et al. (2015) developed a relation between annual duration of overflow occurrence
and rainfall threshold category derived from rain gauge data.  Mailhot et al. (2016) describe
the value of rainfall data in terms of spatial co-location and temporal coincidence, and
mention the limitations of rain gauge data in evaluation of CSO occurrence.  The work of
Mounce et al. (2014) includes radar-rainfall information and focuses on forecasting CSO
occurrence and less on identification of catchment-scale rainfall characteristics.    Both
Montalto et al. (2007) and Lucan and Sample (2015) incorporate scenarios of runoff modeling





using modified forms of traditional design rainfall derived from regional climatology to
illustrate the performance impact of a variety of Green and LID (low impact development)
strategies on CSO occurrence.
This work moves toward understanding rainfall through downscaling in urban areas as
a means to identify specific rain event spatio-temporal characteristics. A occurrence may be
improved by defining fine spatiotemporal resolutions.  The challenge in CSO mitigation
however, is that for most sewersheds, common operationally measured rainfall characteristics
(rain event duration, total rain volume, intensity, continuity (inter-event time, IET),
seasonality and storm type (e.g., stratiform, convective, frontal, orographic, tropical storm
remnants), are determined based on spatially distant point source rain gauges and lack
catchment specific detail. Data records with relevant spatial and temporal details describing
precipitation variability is necessary to define the characteristics of storm events triggering
CSO events.
Historically, rainfall monitoring by ground-level rain gauge networks is considered a
reliable measurement system for many hydrologic applications because it physically captures
pluvial water. In hydrologic engineering and research, rain gauge measurements frequently
serve as a reference for evaluation of indirect or remote sensing rainfall estimation systems
such as weather radars and satellites (Habib et al., 2012; Price et al., 2014; Chen et al., 2015;
Fencl et al., 2015). Several recent studies have utilized existing rain gauge data from single
points or networks. However, the detection of spatial rainfall variation by gauge networks is
limited, in particular at finer temporal resolution. Qi et al. (2016) studied the quality control of
a rain gauge network of hourly data to alleviate error sources. Girons Lopez et al. (2015)
focused on spatial variation of the rainfall for a sizeable watershed (30000 km²). Gianotti et
al. (2013) mentions the limitation of rain gauges use in seasonal weather prediction. Based on





these studies, indications are that both rain gauge location and data quality control are
important. The focus of this study is application of radar and rain gauge data in the small-
scale sewershed and consideration of spatial variation. Additionally, a fine temporal
resolution is maintained to preserve coincidence of rainfall and peak overflow variation and
reduce information loss. Thus, this study includes characteristics of localized rainfall events,
which may vary significantly from spatial averages, through use of radar-rainfall products and
therefore explicitly incorporates rainfall spatial variation.
Hydrometeorologic rainfall monitoring and measurement technology of the weather
radar has advanced in recent decades (Karamouz and Nazif, 2013; Morita, 2011; Yang et al.,
2013). While existing research has been directed to developing Z-R relationships for one-
hour rainfall accumulations (Baeck and Smith, 1998), this study focused on linking radar
reflectivity to rain gauge networks for short duration applications (less than one-hour). By
synchronizing radar rainfall with rain gauge measurements the dependency (and associated
uncertainty) of the Z-R conversion on storm type classification (convective, tropical, east cool
stratiform and stratiform) is diminished. Unfortunately, the Z-R relationship is not typically
calibrated for a particular hydrologic climate or rainfall type and no real-time automated
optimization is implemented (Chumchean et al., 2003; Ice, 2014). Synchronizing radar
reflectivity data with the rain gauge network, a more precise estimate of rainfall (depth,
spatial and temporal variations) indicate improved rainfall estimates at scales of 0.5 $km^2$ and
0.5 hours. In this study, support vector classification (SVC) is used to partition storm events by
underlying characteristics and includes an optimization process for Z-R parameter estimation;
the radar-rainfall data are areal rainfall observations at the sub-hectare (radar polar coordinate
pixel size) resolution at sub-hour temporal intervals.



Identification of rainfall events, for example, using the EPA criteria for urban areas
and a defined inter-event time (IET), provides a context for identifying rainfall spatial and
temporal characteristics associated with CSO overflow events. Accordingly, preparation of
accurate rainfall data, quality controlled weather radar data, identification of independent
rainfall events and corresponding CSO event hydrographs are essential to developing a
quantitative understanding of the phenomenon.
To this end, an objective for this study includes application of locally optimized radar-
rainfall to an urban sewershed (watershed) using fine-scale spatiotemporal resolution data,
and evaluation of rain event characteristics resulting in CSO events. This work generates a
coincident data set of coupled radar-rainfall-CSO events. Categorization of the severe rainfall
events inducing CSO occurrence can provide insights for hydrologic and hydraulic design
guidelines to reduce sewer overflows from combined sewer systems in an urban area.

**2. Local weather radar optimization**
**2.1. Radar rainfall source and preprocessing**
The study region for this work is the city of Louisville, Kentucky (USA) during the
period January 2010 to December 2014. Rainfall data from an operational rain gauge
network, managed by the local utility agency Metropolitan Sewer District (MSD), are the
ground reference values (Hyun et al., 2016). The region's Next-Generation Radar
(NEXRAD) (denoted by call letters KLVX) is located at Fort Knox, Kentucky about 40-km
southwest of the city of Louisville.
In case studies of rainfall spatiotemporal structure, a correlation near 0.6 at 5-km
distance for quarter-hour temporal resolution using ground based rain gauges is shown (Ciach
and Krajewski, 2006; Jung et al., 2014; Mandapaka and Qin, 2013). The average inter-gauge





distance in this application is slightly greater and the gauge network can therefore benefit
from the complementary spatial detail provided by weather radar.  Although the radar data are
not explicitly filtered for error adjustment, the large quantity of data compiled for use, from
gauge network and radar archives, is expected to minimize bias.   Additionally, the proximity
of the study area relative to the radar site, at about 40 km range, is expected to diminish the
influence of common radar error influences such as range effects of signal attenuation,
anomalous propagation, beam blockage, and beam spreading (Ciach et al., 2003; Gorgucci
and Baldini, 2015; Hunter, 1996; Kalogiros et al., 2013; Krajewski and Vignal, 2001; Morin
et al., 2003; Seo et al., 2000; Vignal and Krajewski, 2001).

The fifteen rain gauges of the MSD network are mechanical tipping-bucket type with

resolution of 0.01 inch and temporal interval of five minutes. The data records for radar base-
scan reflectivity (Level II - NEXRAD dual polarization, 0.5 dBZ increment) were retrieved
from the National Climatic Data Center (NCDC) at National Oceanic and Atmospheric
Administration (NOAA). The data cover the entire five-year study period, 2010-2014, in the
format of coded reflectivity volume scans.  The raw reflectivity was converted to a Cartesian
coordinate (ESRI ASCII grid files) system using NCDC's Weather and Climate Tool Kit
(WCT), version 3.7. The WCT provides visualization and export tools to manipulate radar
data.   The Constant Altitude Plan Position Indicator (CAPPI) data, 1-km above the radar
elevation, forms the base-scan reflectivity array over the study area. The spatial pixels are
approximately 220-m square in a Cartesian coordinate grid over the study area, thus pixel area
is less than 5-hectare (15-acre).

The raw radar reflectivity data are instantaneous values and require conversion to

rainfall intensity and accumulation to define volumetric rainfall.  In order to geo-synchronize
reflectivity pixel locations with reference network gauge locations, radar reflectivity pixels



with spatial locations corresponding to MSD gauge locations were identified. In the same
vein, a temporal synchronization was performed to identify radar scan time stamps
corresponding to the local time zone (accounting for daylight savings time as appropriate).
For each rain gauge site, the collocated radar pixel and adjacent eight pixels were identified
for use in data evaluation. Among these nine radar pixels, the single collocated radar pixel
value was selected when reflectivity was within 50% of the average of surrounding pixel
values. Where the difference between the center pixel value and surrounding values was
more than 50%, the averaged reflectivity value was assigned. In the case where a majority of
the 9 radar reflectivity data showed a status or condition as "not available" the pixel status
was defined "not available".

The KLVX radar data management system applies a Z-R relationship according to

four storm types: convective, tropical convective, stratiform, and east cool stratiform.
Reflectivity transforms into rainfall intensity as an instantaneous value, whereas the gauge
values are demonstrate accumulated rainfall over five minutes (Ulbrich and Lee, 1999).
Therefore, instantaneous intensity is further transformed into accumulated rainfall following
application of the Z-R relationship. The first step considers all four reflectivity to rainfall
intensity (mm/hr) conversions. Additionally, temporal synchronization was required since
radar observations are not recorded at equal or uniform temporal intervals. Instead,
reflectivity scan intervals cover a 4- to 10-minute range due to the operational mode of the
weather radar. Generally, three or four instantaneous base scans cover the quarter-hourly
period, and each volumetric scan is weighted according to the inter-scan time interval within
the quarter-hour interval. The process is defined in Equation (1) (Appendix) based on a
weighted time of occupation for each scan within the quarter-hourly interval. For each
reflectivity volume scan, the time interval is defined as the duration from the observation until

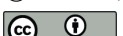



the next observation recorded (inter-scan interval) or the end of the fifteen-minute
accumulation window.

Figure 1 illustrates the volumetric radar rainfall products at monthly to quarter-hour

temporal resolutions. The quality of quarter-hourly radar rainfall estimation is relatively low
while hourly and longer accumulated rainfall products reveal better agreement with gauge
rainfall. However, this study focuses on the shorter duration, quarter-hourly interval, in order
to illustrate radar rainfall products for use in smaller urban catchment applications (Cunha et
al., 2015; Krajewski et al., 2010; Smith et al., 2007; Wright et al., 2014). In Figure 1, most
data are found at the depth range of 5 mm or less for quarter-hourly rainfall because rain
gauge data reliability and detection is sensitive during light rainfall (Ciach, 2002; Humphrey
et al., 1997). For these reasons, a quarter-hour rainfall accumulation threshold of 5 mm was
implemented for evaluating the Z-R relationship in the remaining portion of this study.
**2.2. Radar rainfall optimization by support vector classification**

Application of the optimal Z-R relationship, selected as the one yielding lowest Root-

Mean-Square-Error (RMSE), for each 15-min rainfall accumulation and each rainfall type
category, is summarized in Figure 2 (origin at 5mm rainfall threshold). The rain type
categories corresponding to tropical and east cool stratiform show a dispersed result.
Conversely, the convective and stratiform types tend toward agreement with gauge values as
indicated by the lower variance and narrower clustering along the one-to-one line. Further
optimization processes are considered for the stratiform type rainfall values since the result
shown is considered adequate for this work. In the case of east cool stratiform type, most
rainfall depths are less than 10 mm, and this depth is less significant from a hydrologic runoff
generation perspective. For this reason, the east cool stratiform rainfall type is not considered
in the remaining part of this study. This leaves convective and tropical type rain categories





for consideration, and the focus is on development of an optimization process to improve
agreement of radar-based and gauge-measured rainfall accumulation.

Figure 3 presents a comparison of the convective storm type radar depths and the

gauge rainfall depths.   The lighter shaded markers indicate use of the standard Z-R
transformation with parameters (a:300, b:1.4) and the darker marker dots indicate the
optimized Z-R result.   A simplex optimization procedure was applied to optimize the Z-R
parameters over value ranges of 10 to 500 and 0.5 to 3 for a and b respectively.   Optimization
decreased RMSE and Z-R parameters values of 300 and 1.4 were modified to 250 and 1.4 for
a and b respectively.   The optimized Z-R parameters eliminated the systematic
underestimation but the dispersion is unchanged; the simplex optimization centered all values
about the one-to-one line.

The tropical convective rainfall type contains a large number of high intensity values

and has the widest spread of the comparison groups.   An interesting and challenging issue is
that bias cannot be eliminated by calibration of the Z-R parameters alone.   As shown in
Figure 4 (upper-left), underestimation of rainfall remains following optimization of Z-R
parameters a and b.   In Figure 2 (upper-right, red), the result is shown for the best fit Z-R, yet
the tropical type rainfall appears to form two distinct groups.   The first group is slightly above
the one-to-one line with limited dispersion, whereas a second group is under the one-to-one
line with wider variability. This indicates that a single Z-R relationship for tropical type may
not suffice to encompass the observed characteristics of tropical type rainfall for this region.

In order to investigate a solution for this issue, a SVC optimization procedure was

developed.   The SVC optimization incorporates an unsupervised learning algorithm applied in
the context of a two dimensional surface (x-axis: gauge rainfall and y-axis: radar rainfall).
The concept is a data-based learning process; the SVC creates a linear hyperplane separating



two binary groups according to a separation margin criteria. The hyperplane forms a linear
separation at the maximum margin and is highly efficient in differentiating the non-linear
rainfall characteristics. The determinant in the SVC is a kernel method transformation into a
feature vector (Cristianini and Shawe-Taylor, 2003). In Figure 4 (upper-right), the maximum
instantaneous reflectivity among the group of influential reflectivity values for quarter-hourly
rainfall accumulation defines the kernel. The averaged radar rainfall error (difference from
gauge value) is at the range of 44 dBZ to 47 dBZ and reflectivity of 46 dBZ is selected in
order to balance the number of data values in each group. The kernel method is described in
Equation (2) in the Appendix and defines the two groups through the linear hyperplane. The
similarity function of the kernel method follows training data instead of a fixed set of
parameters, and this feature involves the similarity function k, denoted as kernel in Equation
(2). More simply, the kernel is a weighted sum of similarities between the trained example
input and the new unknown input. The kernel is used as a binary classifier in terms of $\hat{y}$, the
binary classifier for clustering the two tropical groups. In Figure 4 (lower-left) the linear
hyperplane shows two data groups, one group fit with the tropical type Z-R relationship and
the second group in the underestimated region. Following the SVC process, calibration of the
Z-R parameters (RMSE minimum) was completed and the result is shown in Figure 4 (lower-
right). This result demonstrates gains in information for the tropical type rainfall when two
Z-R parameter sets (a, b) are used. The original fan-shaped dispersion is greatly reduced, as
well as the original underestimation issue. The SVC-based solution process provides a multi-
category classification and overcomes limitations of binary classification (Xie et al., 2013).
Use of an uncalibrated Z-R relationship for conversion of reflectivity to rainfall
intensity for short-duration accumulations may result in differences from ground-level rain
gauge observations. An example of these differences is presented in Figure 5 (upper-left)
where results show a fan-shaped spreading with correlation of 0.68 between gauge and radar





rainfall. Coincidentally, underestimation of rainfall may be more problematic than
overestimation in applications of hydrological management and design, and this result is
relevant for the tropical type rainfall category.  In Figure 5 (upper-left), the comparison shows
the best fit standard Z-R relationship (minimum RMSE), and the tropical Z-R relationship
corresponding to the solid red line in Figure 5 (upper-right).  This indicates the tropical Z-R
may not capture a complete description of rainfall variability at higher intensity rainfall rates.
This study introduced an alternate Z-R relationship formed using an SVC optimization
process.  Use of the alternate Z-R relationship produced the results in Figure 5 (lower-left)
and Figure 5 (lower-right).  A comparison of the alternate Z-R (red dotted line) and NWS
tropical Z-R (red solid line) relationships are included. This alternate Z-R relationship,
designated as "tropical-2", is more influential in the higher reflectivity range (greater or equal
to 46 dBZ).  The Z-R relations shown in Figure 5 (lower-right) illustrate the placement of the
existing NWS tropical (tropical-1) Z-R relationship between the convective (black solid line)
and tropical-2 (red dotted line). Graphically, the tropical-1 relationship fills the gap between
the other two convective Z-R relationships.  A notable point is that a large portion of the
quarter-hourly rainfall accumulations occurring at lower radar reflectivity, less than 46 dBZ,
are well represented by the tropical-1 Z-R relationship. The focus of this study on a relatively
shorter, quarter-hourly, rainfall accumulation interval, and the focus on more intense rainfall
values are factors influencing the need to partition this extreme- type rainfall (NWS tropical)
into two sub-groups.

In general, the two tropical sub-groups are similar in the low reflectivity range and

deviate more from one another for reflectivity above 45 dBZ to 50 dBZ.  Based on the use of
categorized Z-R relationships in the region of more extreme rainfall intensity, quarter-hourly
rainfall estimation is improved with a correlation of 0.72 in Figure 5 (lower-left). For




applications in urban hydrologic designs and simulation of historical events, the shorter
temporal resolution of rainfall is useful.

## 3. CSO130 overflow analysis

### 3.1. Urban sewershed setting and CSO location

The sewershed CSO130 is part of the urban CSS and located in an older
neighbourhood, called Buchertown, in Louisville, Kentucky. The specific location of
CSO130 is along Webster Street and its overflow control structure type is a diversion dam.
The sewershed is approximately 13-ha (30-ac) and land-use is a mixture of commercial and
dense residential. The land-use is about 75% impervious with the portions distributed as
residential (24%), commercial (25%), industrial (32%), vacant land (6%), and roads and other
uses (13%). The CSO130 outfall discharges into the nearby stream, Beargrass Creek, which
is a tributary of the Ohio River.
MSD operates a rain gauge network across the city region and one rain gauge is
located near the study area. However, data from this gauge serves only as a reference to
evaluate radar rainfall quality rather than as rainfall for the sewershed. The benefit and
challenge of using weather radar data for operational applications is illustrated and the spatial
variation of rainfall derived from weather radar products described. The radar records are
extracted and optimized, with spatiotemporal resolution of quarter-hour and less than five-
hectare (15-acre), in order to be applicable for the urban hydrologic scale.

### 3.2. Coupled radar rainfall and overflow event record

Only rain events resulting in a CSO event are considered and a quality control
threshold is applied to select rain-overflow events exceeding a ratio of 0.60 for overflow





depth to rainfall depth.  Discriminant analysis is used to categorize the coupled rainfall-
overflow events according to overflow severity; a threshold of the overflow depth of 1.5-mm
partitions the event categories. Results indicate that overflow depth has a strong linear
relationship with rainfall depth and other environmental factors are influential.

Identification of rainfall events, using the EPA criteria for urban areas and a defined

inter-event time (IET), provides the context for identifying overflow events in the CSO flow
record. Accordingly, preparation of accurate rainfall data, quality controlled weather radar
data, and a record of independent rainfall events, are essential.  In radar rainfall estimation,
super resolution data are suitable to define rainfall variation over urban areas.

Figure 6 shows radar rainfall comparisons with gauge rainfall data sources near the

study area. An MSD rain gauge, gauge number TR05, is located about 600-m away from the
study area, and this data record is used as a reference to evaluate spatial variation of rainfall.
The improved radar data shown in Figure 6 (upper-right) has improved the correlation to 0.79
compared with a 0.70 correlation in the original estimate (Figure 6 upper-left).  Two other
rain gauges, TR12 and TR03 show correlation of 0.55 and 0.05 with the TR05 data
respectively.  Within a distance of 15-km, rainfall is spatially uncorrelated ($\rho$=0.05), and
correlation decreased to 0.55 within 5km distance at TR12.  This reveals the high spatial
variation of the rainfall at quarter-hourly temporal resolution and the benefit of radar-rainfall
over the limitation of reliance solely on ground-based rainfall measurement.

The CSO mechanism in the sewershed is not only related to the rainfall characteristics;

depth, duration and intensity, but also the continuity of the rainfall event. Figure 7 illustrates
the extreme overflow cases at CSO130 in time-flow manner and the hydrologic response is
related to rainfall variations within the rain event. The nine overflow events shown indicate
that most were triggered immediately following the heaviest rainfall interval.  Naturally, the



rainfall volume is the primary influence on overflow amount, but it is not the only factor. The
more sizeable rainfall peaks affect the overflow amount and time distribution.  For example,
the sixth greatest overflow in Figure 7, with overflow amount of 10.68-mm, has precipitation
duration less than an hour but the overflow was significant because of high intensity rainfall.
On the contrary, the overflow event which ranked in fourth has no clear intense rainfall
observed; instead rainfall is steady and uninterrupted. These results indicate rainfall event
continuity as another factor triggering overflow event occurrence.
One definition of a rainfall event is provided by the EPA for rainfall event in the context
for urban regulatory settings (Driscoll et al. 1989).  The EPA document defines a rainfall
event as "A minimum storm volume of 0.1-inch (2.54-mm) was specified for the analysis that
were performed, so that the analysis would produce statistics of 'runoff producing' events
within 6 hours." In short, a single rainfall event is completely independent if no sizeable
rainfall, greater than 2.54-mm (0.1-inch), occurs within six hours. The rainfall event defined
by EPA regulation and corresponding time for the overflow event were determined from the
time rainfall began until six hours from the end of the rainfall event. By EPA definition, a
rainfall event is followed by at least a six-hour dry period and so the implied time available
for overflow to occur is limited to six hours following the rain event. Based on this, there are
95 rain events with coupled CSO occurrence in the sewershed for CSO130 over the three-year
study period, January 2011 to December 2013.

## 3.3. CSO130 overflow analysis

The CSS CSO130 control structure is a 0.61-m (24-inch) circular brick sewer pipe
flowing with an average of 12 overflow incidents (events) per year (averaged 2.33 hours of
duration and 90,000 gallons of combined sewer per incident) (MSD, 2014).



### 3.3.1. Quality control of coupled rainfall/overflow event

The number of CSO events identified directly from data records of rainfall and overflow analysis indicates the number of incidents is 95, and this is a greater number than the 35 to 40 otherwise expected according to the average number of reported incidents over the same three-year study period (MSD 2104). Potentially, the method of identification and event partitioning (IET) may influence the number, but proper quality confirmation is required for data reliability. To this end, a means of screening outliers and poor quality data records is utilized. Application of a common rainfall-runoff index threshold screening, based on watershed characteristics, is not possible since the total runoff for each CSO event partitions flow into two directions, one part is the overflow and the remaining portion continues to the water treatment plant. During an overflow event it is not possible to separate the portion attributed only to stormwater runoff. To address this issue, CSO event data are partitioned into acceptable and non-acceptable clusters. Figure 8 (left) shows the normalized runoff-rainfall index ratios of overflow depth to radar-rainfall depth and overflow depth to gauge rain depth for each event. The plot spreads in a two-dimensional field; with x-axis: ratio for rain gauge MSD TR05 (600m from CSO130), and y-axis: ratio for radar rainfall. A ratio greater than 1 indicates runoff greater than rainfall, and this unlikely occurrence may indicate data error or other issues; for this reason, these data are excluded from the study. The use of two rainfall sources lessens the uncertainty concerning rainfall occurrence and incorporates both these practical hydrologic observations into this study. Figure 8 shows a notable absence of overflow occurrence between ratio values of 0.60 and 0.80. Therefore, a threshold ratio of 0.60 was defined as the acceptable coupled rainfall and overflow event criteria; this is the boundary where data are densely populated and shown as the inner region defined by a bold solid line forming a quarter-circle in Figure 8 (left). The right portion of Figure 8 shows all CSO events and the bold solid line indicates a value of 0.6 for the radar




overflow ratio. The result identifies two groups of CSO events: acceptable (blue) and non-
acceptable (red). This process indicates 52 coupled rain-overflow events and this corresponds
well with the expected number as suggested by the MSD report for the three-year study
period (MSD, 2014).

**3.3.2. Analysis of coupled rainfall and overflow events**

**3.3.2.1. Overflow relation to rainfall depth, intensity and duration**

The coupled rain and overflow record for CSO130 shows the sewershed runoff
response is prompt with a hydrograph form similar to a smoothed and time-lagged reflection
of the hyetograph. That being so, understanding the rain event characteristics provides
insights into the timing, intensity and amount of overflow. Fundamentally, the quantitative
relation between rainfall and overflow has a visible linearity as in Figure 9 (left). Rain
volume is an important factor and shows a linear relationship with overflow. As shown in
Figure 9, when rainfall is less than about 8-mm a low overflow volume occurs and overflow
amount increases linearly above this rainfall depth. For overflow values above a 0.40 ratio of
rainfall volume the sewer overflow volume is more significant and likely to impact
environmental quality in the Beargrass Creek. It is expected that total rainfall depth is a
significant factor triggering an overflow, but this simplified conclusion cannot completely
explain the behaviour and a search to understand the contributing factors causing overflow
events is warranted.
The rain event duration and peak intensity (15-minute temporal resolution) are
important in determining overflow volume in Figure 9 (right). In the duration versus rainfall
depth field (Figure 9 right), the events clearly divide into two groups when clustered by peak
rainfall intensity. The two groups have somewhat different tendencies in the two-dimensional




space with the strong peak intensity group showing a relatively short duration but larger
volume of rainfall.  On the other hand, rainfall volume tends to be relatively stable and less
relevant as an overflow trigger no matter the event duration.  In the small-scale urban
watershed setting, existence of high-intensity peak rainfall may produce significant
volumetric rainfall, thus, rainfall intensity significantly impacts drainage system performance
in urban areas (Arnbjerg-Nielsen et al., 2013; Mamo, 2015).  Considering this result in a
practical application, rainfall depth, intensity and duration are all factors indirectly
incorporated into historic intensity-duration-frequency (IDF) curve used to define volumetric
rainfall for urban hydrologic design.  However, variations between rainfall observations and
IDF design values illustrates the uncertainty for applications requiring fine spatiotemporal
resolution such as urban sewersheds where runoff response occurs well under the sub-hour
temporal scale.
**3.3.2.2 Overflow relation to rainfall depth, storm type and continuity of rain**

There is a thread of inter-connection between instantaneous heavy rainfall, storm type

and resulting overflow in this small watershed. A concurrence of rainfall characteristics and
watershed condition, for instance, existing antecedent moisture, wet or dry surfaces and soils,
which effect rainfall retention and percolation, may influence overflow occurrence.
Therefore, rainfall continuity in terms of single events is considered as an additional factor.
The extreme overflow events in Figure 7 reveal the importance of the rainfall continuity since
there are four high ranked overflows (rank number: 3, 4, 8, and 9) associated with rainfall
events with relatively insignificant peak intensity (below 5-mm/15minute), but continuous
and uninterrupted rainfall.  In other words, the length of the duration of rainfall within a single
event is an influencing factor related to the CSS capacity and resulting CSO for this
sewershed.



The ratio of the time rain falls during a rain "event" to total event duration represents the continuity of the rainfall or rainfall occupancy ratio. Figure 10 (left) illustrates the relationship between rainfall depth and rainfall occupancy by rain type and season; warm season (April to September) and cold season (October to March). The radar rainfall product indicates rainfall type for each 15-minute rainfall accumulation. The characteristics of a single storm, in motion over CSO130 sewershed, are dynamic and a series of storm cells may move over the area. The convective storm type may have a single or multiple cells within the developed storm structure associated with severe rainfall (Caine et al, 2013; Cetrone and Houze, 2006; Feng et al., 2014; Peter et al., 2015; Zawilski and Brzezińska, 2014).

Identification of rainfall type is based on the ratio of number of convective type radar pixels to total rain pixels in the storm. Applying a threshold ratio of 0.45 results in the two event groups; a convective prevalence group and a stratiform prevalence group. The stratiform group has no discernible spatial pattern features other than the continuity of rainfall coverage, while the convective group has a tendency toward increasing intensity beginning around a ratio of 0.60. The highest three rain depth events are in the convective group. The reappearance of the rainfall overflow plot (Figure 10, right), with seasonal rainfall group details added, demonstrates the characteristics of the overflow inducing rainfall events. Prior to presentation of this figure, the nine ranked overflow inducing rainfall events show a 0.81 ratio of rainfall occupancy and no event with lower than a 0.60 ratio. This indicates the rainfall event group most likely to generate a CSO are the convective rain group in summer season. The mitigation of combined sewer overflow events can use this information in hydrologic design to improve future approaches to stormwater runoff reduction. The overflow of CSO130 is a response to the interaction of natural rainfall variability, the urban landscape, land-use and hydrologic environment. In addition to rainfall variability, other qualitative factors influence the likelihood of overflow occurrence. Therefore, understanding the




temporal and spatial structure of overflow inducing rain events is useful to estimate overflows
in CSOs.

### 3.3.2.3 Discriminant analysis in overflow inducing rainfall events

This study shows that an overflow event in a sewershed is induced through the
integration of factors from two fields; natural rainfall variability and the constructed
sewershed conditions. The fundamental assumption is that rainfall induces the overflow event
in a small-scale sewershed because the runoff response is rapid and the hydrograph structure
resembles the hyetograph. Thus, preventing overflow events inevitably requires
understanding of rainfall characteristics. The volumetric rainfall depth was shown to relate
linearly with overflow and other factors, such as rainfall intensity, duration, and continuity of
rainfall (as a ratio of rainfall occupancy) as influential factors.  Discriminant analysis is
introduced to classify these precipitation factors. The discriminant analysis uses the
combination of features from the continuous independent variable (rainfall characteristics) to
define a separation of the discrete dependent variable (Martinez and Kak, 2001; Tahmasebi et
al., 2010) and is applied broadly in water resource (Sangam et al., 2008, Boyacioglu, 2010).
In order to apply discriminant analysis, the dependent variable (overflow) must be categorical
unlike the continuous independent variables. In Figure 11, a threshold overflow depth set at
1.5-mm, for the CSO130 sewershed, and categorizes the coupled rainfall-overflow events into
two groups; a significant overflow group (23 events referred to as group-1) and non-
significant overflow group (29 events referred to as group- 0).
In Table 1, the mean values of the rainfall related variables influencing the overflow
events are presented in discriminant groups.  As expected, this highlights differences between
groups and provides a quantitative distinction of the decisive overflow factors. The mean
rainfall depth is 8.85-mm in the non-significant group (group-0) and 23.70-mm in the





significant group (group-1). Overall, volumetric rainfall governs the overflow in this small
scale sewershed. The peak rain intensity has a similar tendency showing 3.76-mm per 15-
minute and 6.20-mm per 15-minute for group-0 and group-1, respectively. Commonly, the
rainfall depth and the rainfall intensity (peak) have positive correlation with overflow amount.
However, the duration of rainfall indicates a contrast to this expected result. In Figure 5
(right), the majority high rainfall depth events have shorter durations. These shorter rainfall
duration events are expected to fall into group-1 considering the relationship between rainfall
depth, intensity and overflow occurrence. In this case, the continuous, uninterrupted and
longer duration rainfall induces the overflow.  This is due to the inverse correlation between
rainfall duration and rainfall continuity, where a correlation of -0.64 is indicated between
these independent variables.  A longer rainfall duration is more likely to also contain a non-
rain period resulting in proportionally lower continuity.
Another matter that merits mention is the definition of rainfall event duration. The
study incorporates the USEPA (EPA, 2004) definition for continuous rainfall and independent
event identification. The emphasis is on whether event independence, using the temporal
separation of a six-hour dry period for a small-scale sewershed, is appropriate since both the
sequence of rain depth and continuity of rainfall within the rain event are influencing factors.
The differences are investigated here using discriminant grouping by rain type and rainfall
continuity, nevertheless overflow occurrence is associated more with rain events in group-1.
The definition of a rainfall event may be improved with incorporation of factors such as
watersheds size, land-use characteristics, and hydrologic goals.
Table 2 shows 78.8% (41 of 52) rainfall events are correctly classified using the linear
combinations identified by discriminant analysis. Under the predefined threshold overflow
depth of 1.5-mm, the 29 and 23 coupled events fall into non-significant and significant groups



respectively. This threshold considered a balance for the number of events in each group. The
objective discriminant group clustering indicates 12 events in the significant group and 40
events in the non-significant group. The linear combination of rainfall factors, $\vec{w} \cdot \vec{\mu}$, include
the mean and variance for the clustered factors. The cluster grouping decision includes the
ratio of variances within and between the groups as defined by equation (3) (appendix).
Based on this formulation, each group was established by the lowest variance of rainfall
factors. The corresponding overflow depth is found to be about 4.6-mm for a rain event with
24-mm depth (single rain event category).
This type of information, specifically identifying overflow volume associated with
rainfall event characteristics, may serve as an indicator of overflow potential in a CSO
sewershed. Information defining a gradation of overflow potential may be useful for
operational optimization such as, real-time likelihood of an overflow occurrence, design of
overflow dam height or pipe size, or design of retro-fit infrastructure to mitigate significant
overflow events. In this study, CSO occurrence in a small-scale sewershed is investigated
with a focus primarily on rainfall characteristics.
**4. Conclusions and Future works**
Improving and preserving water quality and the aquatic environment in urban areas is a
focus of the EPA and a component of regulatory guidelines limits the allowable occurrence of
CSO (combined sewer overflow) events (EPA 1994, EPA 2004). The approaches for
mitigating overflow events require information to define existing CSO conditions and event
occurrence in terms of flow volume, seasonal variation, and pollutant type and concentration.
Use of an uncalibrated Z-R relationship for conversion of reflectivity to rainfall intensity
for short-duration accumulations may result in differences from ground-level rain gauge
observations. Coincidentally, underestimation of rainfall may be more problematic than


overestimation in applications of hydrological management and design, and this result is
relevant for the tropical type rainfall category. This indicates the tropical Z-R may not capture
a complete description of rainfall variability at higher intensity rainfall rates. This study
introduced an alternate Z-R relationship formed using an SVC optimization process. This
alternate Z-R relationship, designated as "tropical-2", is more influential in the higher
reflectivity range (greater or equal to 46 dBZ). A notable point is that a large portion of the
quarter-hourly rainfall accumulations occurring at lower radar reflectivity, less than 46 dBZ,
are well represented by the tropical-1 Z-R relationship. For applications in urban hydrologic
designs and simulation of historical events, the shorter temporal resolution of rainfall is
useful.

The volumetric approach of CSS overflow event study in a small-scale sewershed is

presented using the radar-rainfall characteristics. The study incorporates details of radar-
rainfall data evaluation, rain event definition, and reveals the dependency of CSS overflow
events on rainfall depth, duration, intensity, type and continuity. The radar derived rainfall is
necessary to determine rain depth over the region of interest where the coverage of rain
gauges is limited.

Fundamentally, a linear relation exists between rainfall and overflow depths governing

the occurrence of CSO events in this small-scale sewershed. The identification of
corresponding rainfall and overflow events requires evaluation of coupled rain-overflow
events and the study determined an overflow depth to rainfall depth ratio of 0.60 indicative of
valid events.   The discriminant analysis clustered overflow events into overflow severity
classes. The objective classification categorized most events correctly and the discriminant
analysis provided an indication of the volumetric relationship between the rainfall and
overflow in this sewershed system.





The more significant sewer overflow events occur rapidly, typically within a few hours
following rainfall and from rainfall event durations less than a half day.  This means daily or
longer rainfall records may not be suitable for overflow analysis for small-scale sewersheds.
This is in part due to the lack of independence in the identification of overflow inducing rain
events where rain intensity, rain continuity and variability definition within a single rain event
are necessary.  An in-depth investigation of rainfall and overflow relationships across a range
of hydrologic settings and sewershed characteristics may reveal an index for the practical
design of a sewer overflow prevention structures. This type of study is essential for optimal
development of objective and quantitative methods to mitigate CSS overflows in urban
environmental systems.

**Appendix: Mathematical background**
Equation (1)
$$\mathbf{R_Q(t,x)} = \sum_{i=1}^{n} \left(\frac{\mathbf{R_i}}{\mathbf{4}}\right)\left(\frac{\mathbf{t_{i+1} - t_i}}{\mathbf{2}}\right)$$
where,
$\mathbf{R\_Q\ (t,x)}$ is weighted accumulation of quarter hourly rainfall capsule at the fixed location where
reference data are corresponded in arbitrary time, t
$\mathbf{n}$ is influential number of reflectivity values for  $R\_Q\ (t,x)$ which fall into the quarter hourly
capsule and front and the rear reflectivity value when it is influential to the capsule.
$\mathbf{R\_i}$ is converted rainfall rate in millimeter per hour among four different NWS Z-R relationships
$\mathbf{t\_i}$ is time of observation of instantaneous radar base scan
Equation (2)





$$\hat{y} = \mathbf{sgn} \sum_{i=1}^{n} \mathbf{w_i y_i k(R_i, \acute{R})}$$
where,
$\mathbf{R_i}$ and $\mathbf{y_i}$ are instance-based learners for the i-th training example
$\mathbf{w_i}$ is a corresponding weight
$\acute{R}$ is an unlabeled input
k is a function of the weighted sum of similarities which is called kernel
$\hat{y}$ is a prediction label of kernelized binary classifier
Equation (3)
$$T = \frac{(\vec{\mathbf{w}} \cdot \vec{\mu}_{Significant} - \vec{\mathbf{w}} \cdot \vec{\mu}_{Non-Significant})^2}{\vec{\mathbf{w}}^T COV(R_{i,j})_{Significant}\vec{\mathbf{w}} + \vec{\mathbf{w}}^T COV(R_{i,j})_{Non-Significant}\vec{\mathbf{w}}}$$
where,
$\mathbf{T}$ is a decision factor for the clustering of Significant and Non-Significant groups of the overflow
event
$\vec{\mu_i}$ is mean value for the clustered group.
$R_{i,j}$ is rainfall variables among rainfall depth, duration, intensity, type, continuity.

**Acknowledgements**
This sewershed study was partially supported by the MSD 'Green Infrastructure' project as a
co-operative study with the Center for Infrastructure Research at the University of Louisville.
We appreciate the reviewers for comments and suggestions that lead to an improved
manuscript.



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





**Tables**
Table 1. Group Mean Values of Rainfall Characteristics by Discriminant Analysis.

| Groups | Variables | Mean |
|---|---|---|
| Non-Significant Overflow (group 0) | Duration (hour) | 4.57 |
| | Rain Total (mm) | 8.85 |
| | Rain Peak (mm/15min) | 3.76 |
| | Rain Type (convective ratio) | 0.47 |
| | Rain Continuity ratio | 0.56 |
| Significant Overflow (group 1) | Duration (hour) | 6.14 |
| | Rain Total (mm) | 23.7 |
| | Rain Peak (mm/15min) | 6.20 |
| | Rain Type (convective ratio) | 0.50 |
| | Rain Continuity ratio | 0.60 |
| Total Events | Duration (hour) | 5.37 |
| | Rain Total (mm) | 15.4 |
| | Rain Peak (mm/15min) | 4.84 |
| | Rain Type (convective ratio) | 0.48 |
| | Rain Continuity | 0.58 |





Table 2. Classification Result and Predicted Group Membership by Discriminant Analysis.

| Overflow Severity | | | Predicted Group Membership | | Total |
|---|---|---|---|---|---|
| | | | Non-Significant | Significant | |
| Original | Count | Non-Significant | 29 | 0 | 29 |
| | | Significant | 11 | 12 | 23 |
| | Percentage | Non-Significant | 100 | 0 | 100 |
| | | Significant | 47.8 | 52.2 | 100 |


















**Figures**

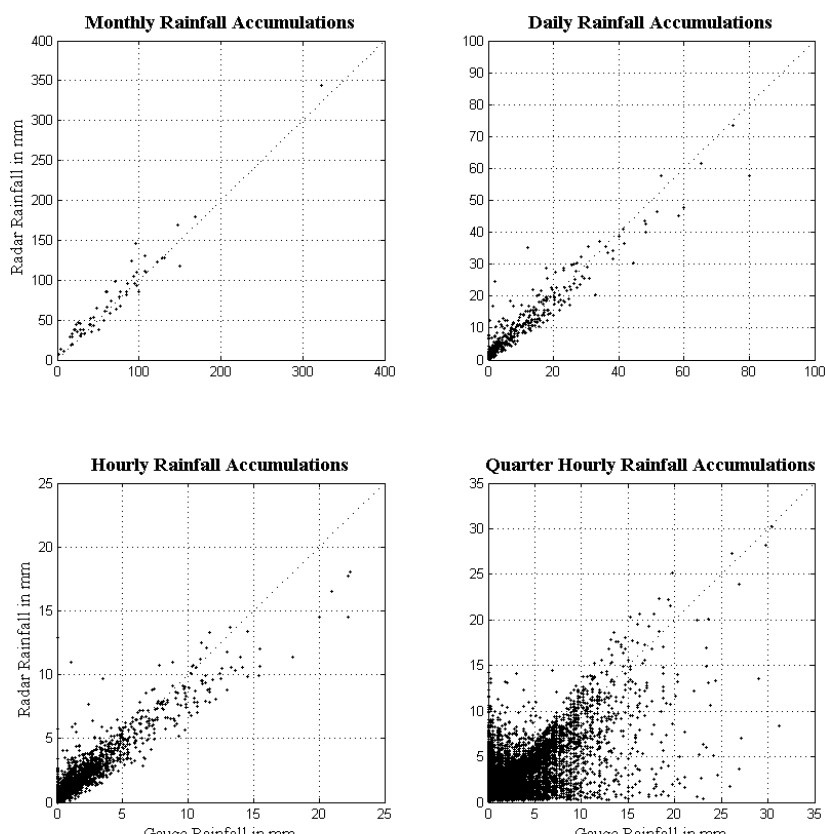


Figure 1. Gauge and Radar rainfall depth in 2-dimensional space (gauge-radar volume) across
temporal resolutions: monthly (upper- left), daily (upper-right), hourly (lower-left), quarter-
hourly (lower-right)

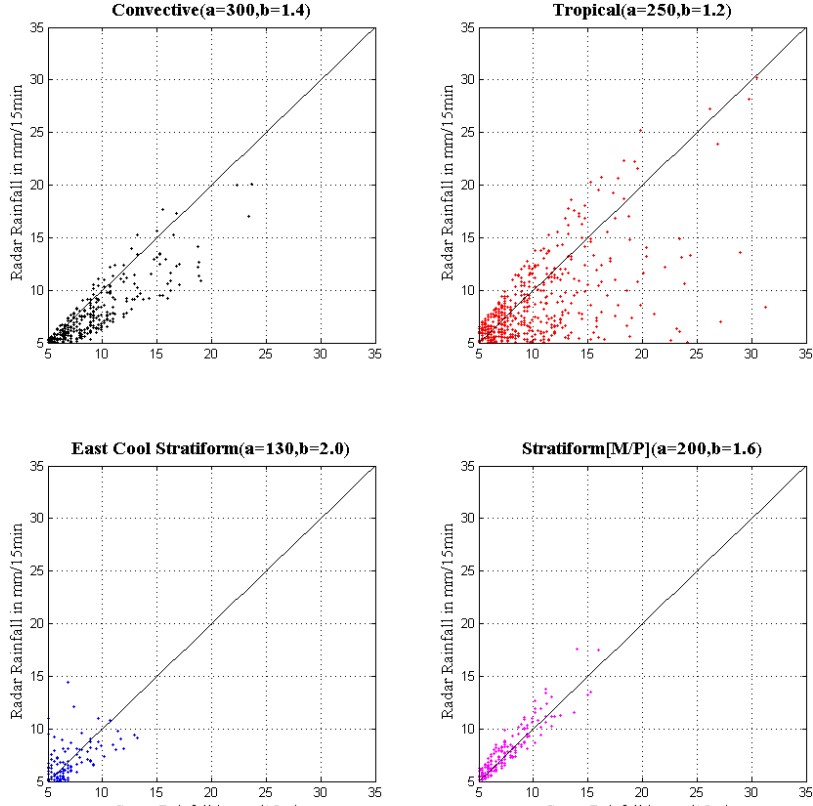


Figure 2. Scatter plots of rainfall volumes for each storm type after optimization (minimum
RMSE error): convective type (upper-left), tropical type (upper-right), east-cool-stratiform
type (lower-left), stratiform – Marshall/Palmer - type (lower-right)





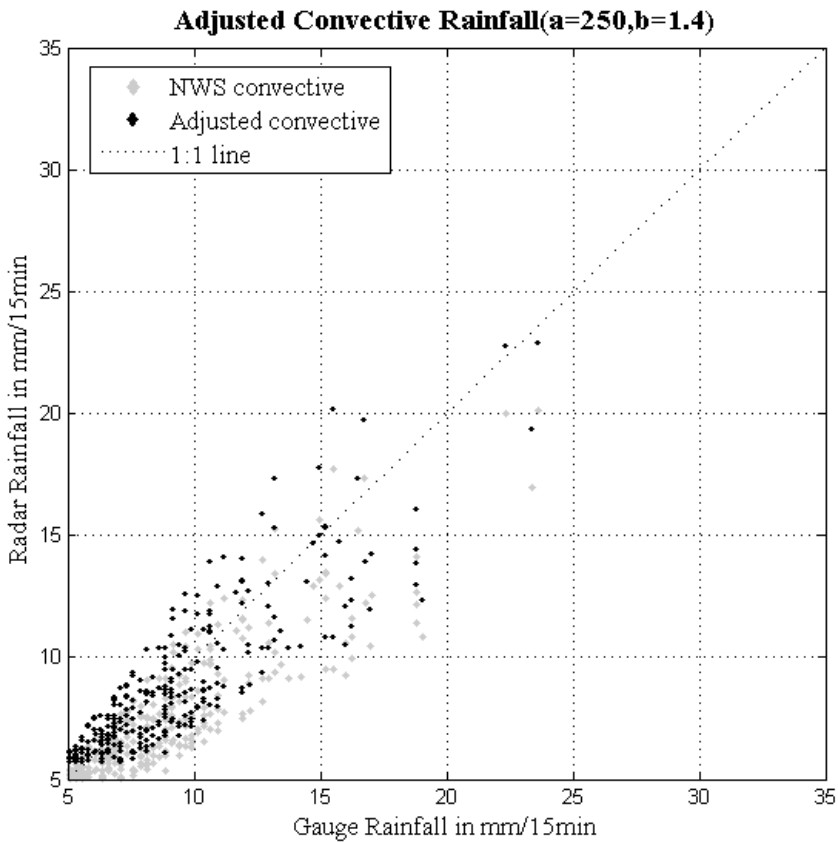


Figure 3. Convective rainfall type: radar and gauge comparison with (a) standard NWS Z-R
relation (light shade marker), and (b) optimized Z-R relation (minimum RMSE error)



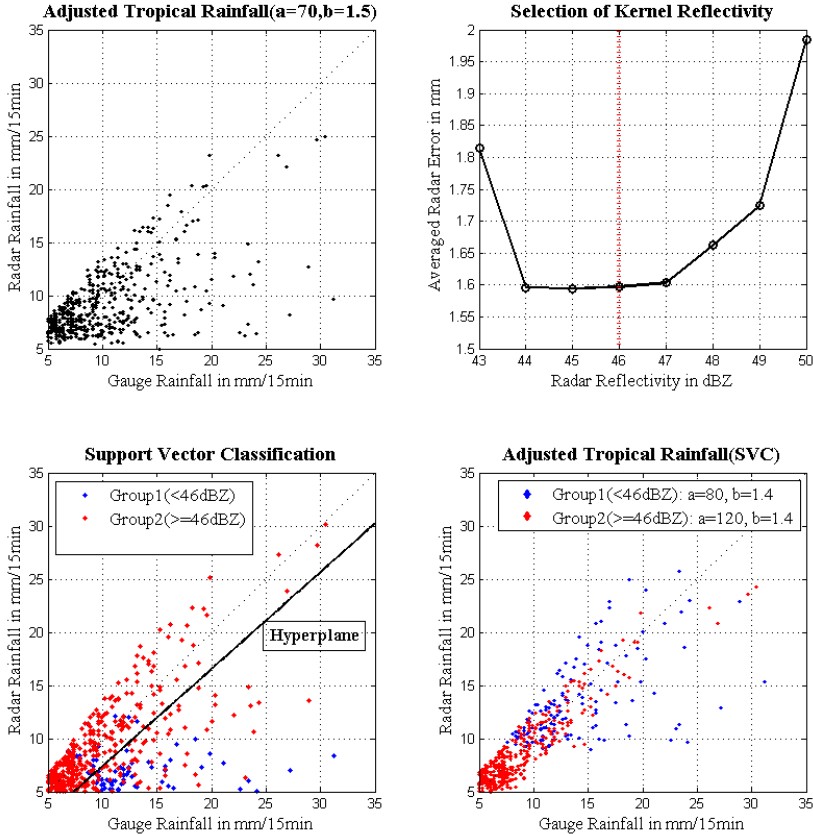


Figure 4. Tropical type rainfall results: Optimized minimum RMSE error (upper-left);
Decision schematic for SVC kernel within least RMSE error range (upper-right); SVC binary
clustering hyperplane (lower-left), SVC-based optimization with two rainfall groups (lower-
right)





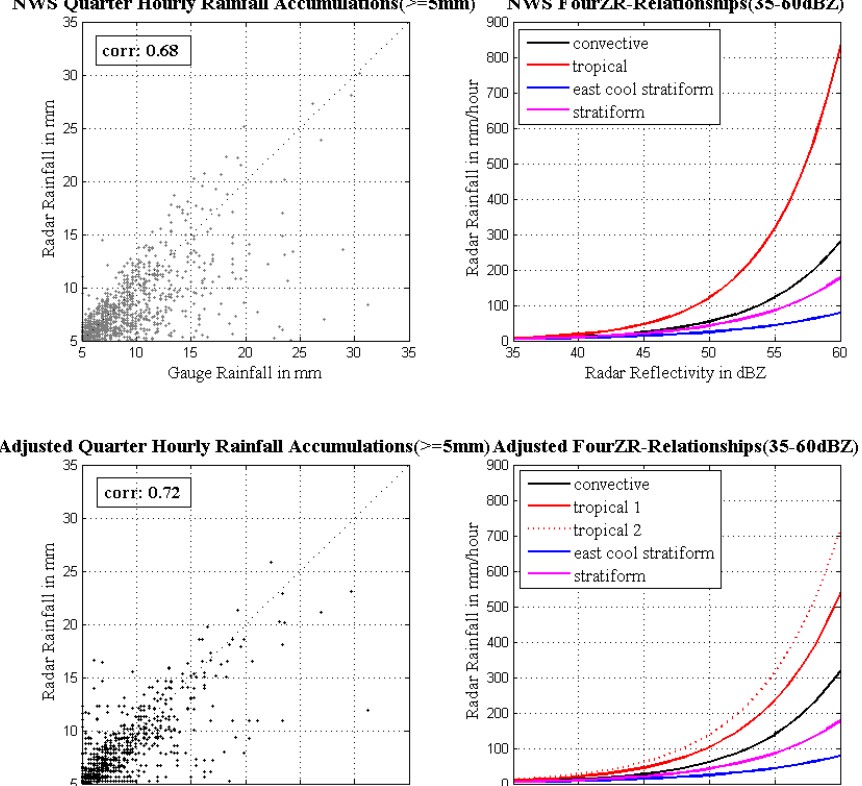


Figure 5. Comparison of local radar rainfall estimations and corresponding Z-R relationships:
(NWS Standardized Z-R-based quarter-hourly rainfall accumulation (upper-left); four
empirical NWS Z-R relationships (upper-right); Optimal SVC-based quarter-hourly rainfall
accumulation (lower-left); SVC-based optimal Z-R relationships (lower-right) (adapted from
Hyun et al., 2016c).





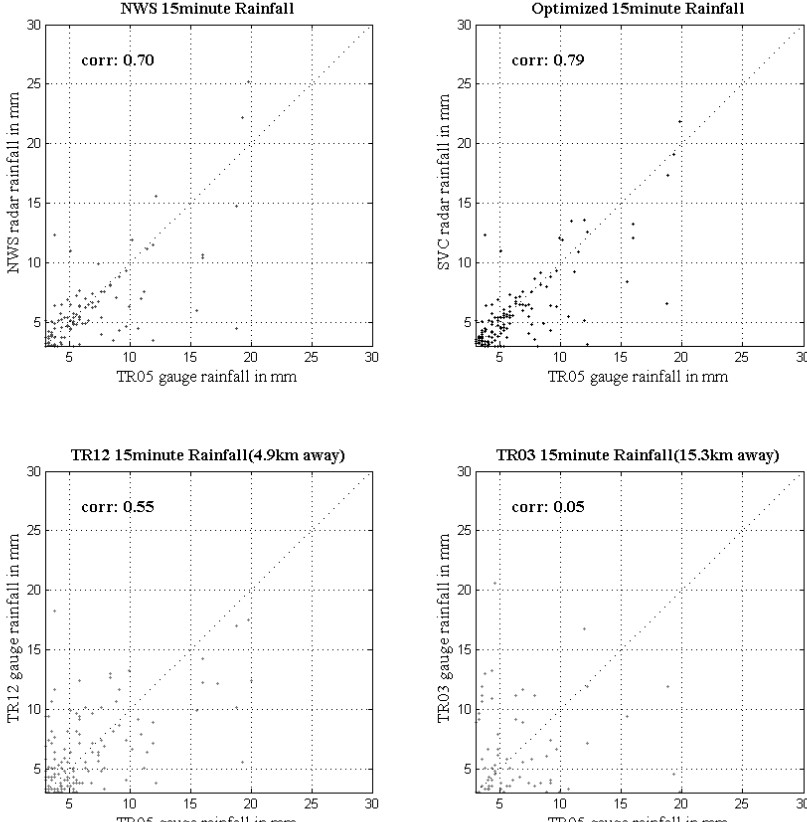


Figure 6. Rainfall data quality comparisons with the reference rainfall data (TR05): NWS
radar data (upper-left), Quality-improved radar data by SVC (upper-right), MSD rain gauge-
TR12; 4.9km away from the study area (lower-left), and MSD rain gauge-TR03; 15.3km
away from the study area (lower-right).




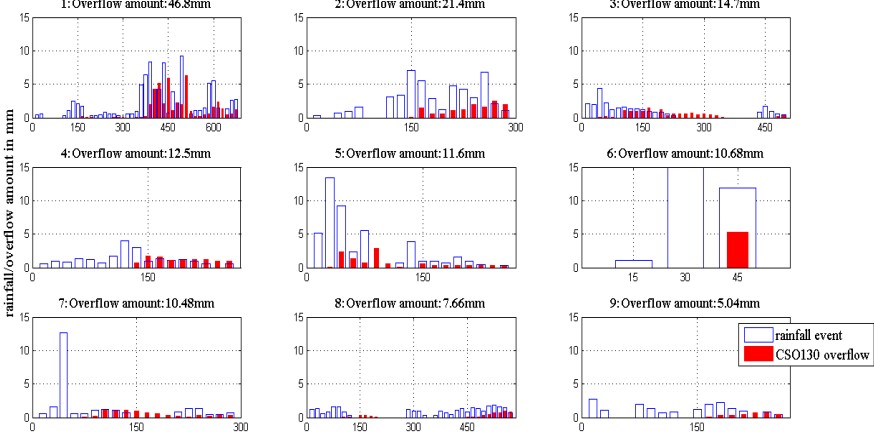


Figure 7. Selected CSO events and corresponding rainfall events; Event number denotes the
rank of the overflow amount through the outfall structure to Beargrass Creek, Louisville, KY.

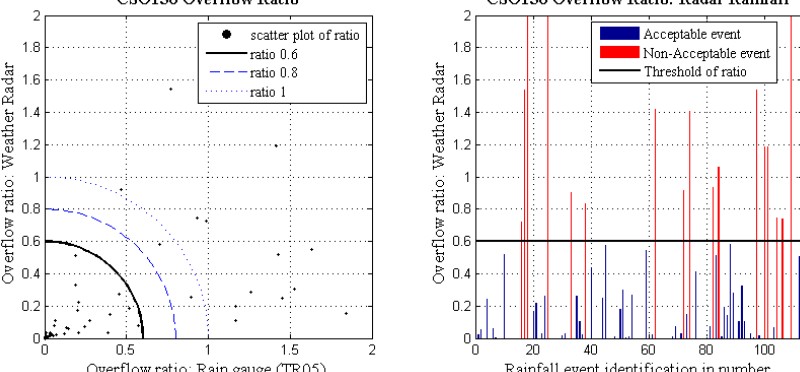


Figure 8. Overflow ratio plots. Left side: Two-dimensional radar & gauge rainfall field; x-axis
shows rain gauge ratio-MSD TR05 (nearest study area), y-axis shows radar ratio-NWS
weather radar KLVX.  Right side: Criteria threshold for valid event selection: 52 acceptable
events (blue) below the 0.60 overflow/rainfall threshold, and 43 non-acceptable events (red)
exceed the threshold.







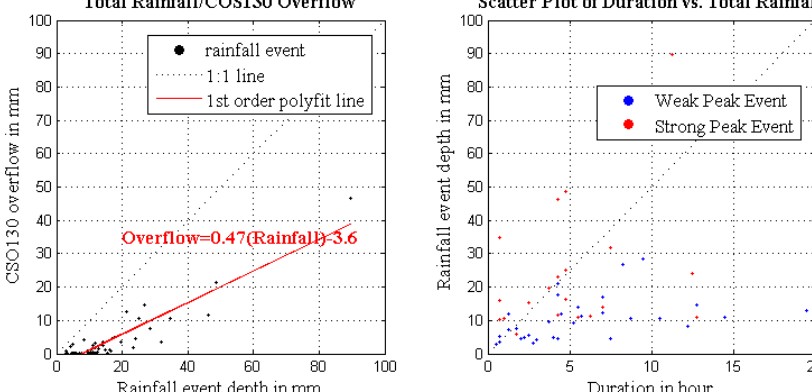


Figure 9. Event-based rain depth versus overflow depth (left), and rain event duration versus
rainfall depth grouped by peak rain intensity (right).    Intensity threshold peak is
4.87mm/15minute to identify weak (blue) and strong (red) peak event groups.

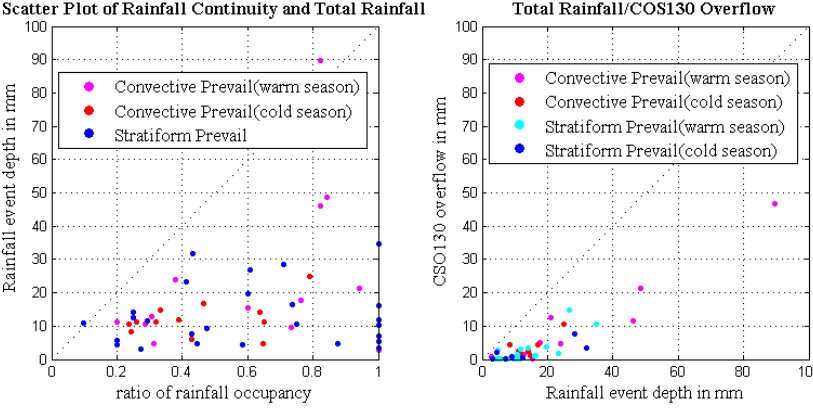


Figure 10. Rainfall occupancy ratio (ratio of continuous rain duration to total event duration)
and total rainfall event depth: convective event type in warm season (magenta), convective
type in cold season (red) and stratiform (blue) (left).    Event-based rainfall depth versus





overflow depth: convective-warm season (magenta), convective-cold season (red), stratiform-
warm season (cyan), and stratiform-cold season (blue) (right).

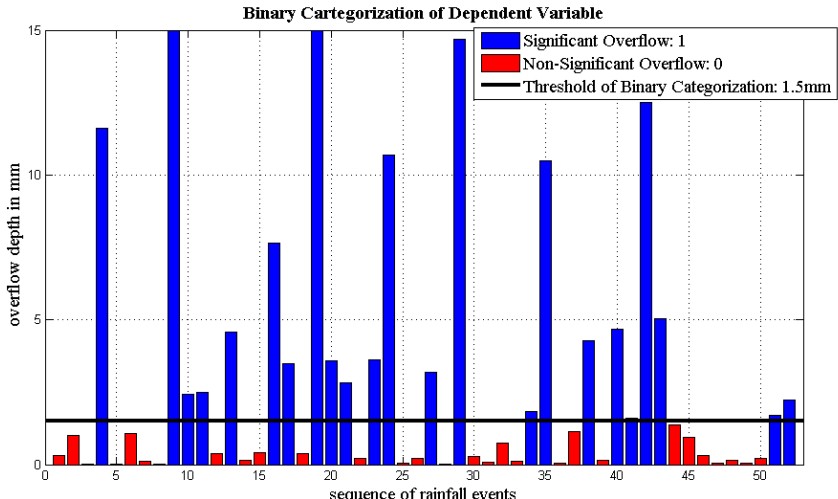


Figure 11. Overflow event and 1.5-mm depth threshold separating overflow events into two
binary categories – significant (denoted as 1) and non-Significant (denoted as 0).