# Peer review of "Manuscript under review for journal Hydrol. Earth Syst. Sci."

_Hydrology and Earth System Sciences, 2016_

## Referee Comment (RC1) · Anonymous Referee #1 · 30 Sep 2016

This paper addresses two interesting and linked questions – first, can we optimise the Z-R relationship for rainfall measured on a short time-scale? Second, what rainfall events cause CSOs? The research is then divided into two parts. In the first part, a Support Vector Classification approach is used to improve the correlation of radar-rainfall measurements by identifying a new Z-R relationship. The authors use a variety of techniques to study these questions and the methods generally are sound. First of all, the paper is generally well written, with only a few typos or sentences that could be improved. However, perhaps the fault rests with me and my reading of the manuscript but there are places where it is not clear what is meant and what precisely was done. This is particularly the case in relation to optimization. I ask questions below merely to ascertain whether my reading of the manuscript is correct. Line 66 – Mailhot 2015, not Mailhot 2016 (unless there is a missing reference) Line 98 "The focus of this study is the

application . . . " Line 104 ". . . with (or by) weather radar . . ." Line 125 "phenomena" (if plural is what is intended) Line 141 – Please specify more clearly what the correlation is here – do you mean the spatial correlation? Line 144 – What is the intergauge distance? Line 174 – Could you justify this step a little further. If the central cell (of the 9) is smoothed to the average, is this not likely to remove what might be heavier rainfall? Alternatively this step could increase low rainfall values. Could you say how often this smoothing took place and how it may have affected the results. Could this be related to the underestimation you see, especially for tropical rainfall? Line 178 – could you explain how the rainfall type is identified? Was this provided to you as part of the original dataset and the classification made by the providers of the data? Or is this what you mean by optimization. (As I have read further, I see on line 436 that this is discussed. Could you explain this earlier in the paper please?) Line 189 – why is this equation in the appendix? Line 205 – Could you clarify what is meant by optimization here? Is it the selection of one of four Z-R relationships, or the modification of the parameters of the Z-R relationship? Line 316 – You talk quite early on in the paper about continuity, but it is not until line 432 where you define it. Could it be defined earlier? Line 319 – here you say super-resolution data is important to the estimation of CSO, but with your smoothing operation (already discussed), you may be losing the high resolution data that is important. If I understand correctly, the pixel resolution is 5 ha. If you use the smoothed result from 9 pixels, you are using the rainfall from an area of 45 ha. Is this correct? Line 370 – I'm not sure I follow the analysis of acceptable and unacceptable overflow events. Where there is an overflow and the overflow ratio is less than 0.6, is anything known about the volume of overflow that reaches the receiving body? What is an acceptable overflow? If I proceed to line 400, we read that for values greater than 0.40, there is a likelihood of significant pollution. Why then is 0.60 chosen as the threshold for acceptable overflow volumes? It is clear from Figure 11 that the unacceptable events are those events with greater volumes, but I would appreciate a better understanding of how you derive at this from the index. Line 389 – could you say something about the relationship between raingauge and radar raingauge for

the CSO causing events. Are they in anyway different from the other rainfall events? Conclusions - Could you discuss any differences you may have seen between the two types of tropical storms – can the information in your research be used to identify in real-time which Z-R relationship to use, and whether it can be used to improve the prediction of CSOs? The conclusions section could be improved. You say early on that "Categorization of the severe rainfall events inducing CSO occurrence can provide insights for hydrologic and hydraulic design guidelines to reduce sewer overflows from combined sewer systems in an urban area". Can you say a little more about how this might be done? Could you also say more about the number of false negatives in Table 2? Out of 52 events listed, you predict 11 out of 52 incorrectly? What is different about these events. If you look at Table 1 – the major differences seem to be rainfall intensity and total depth (the other differences aren't great. This is what you would expect. When do you make a false prediction?

---

## Author Comment (AC1) · 4 Oct 2016

Reply RC1(Anonymous Referee #1) on September 30th 2016

Urban sewershed overflow analysis using super-resolution weather radar rainfall (Manuscript Number. hess-2016-362)

Authors: Jin-Young Hyun, Thomas D. Rockaway and Mark N. French

This document provided detailed response to referee comments (RC1) from Anonymous Referee #1. The authors recognize and thank this reviewer for the effort and suggestions to improve this manuscript.

NO Referee #1 Questions/Suggestions and Addresses

1 Q. Line 66: Mailhot 2015, not Mailhot 2016 (unless there is a missing reference)

Answer. This is a typo and the cited paper was published in 2015 as referenced. The sentence in lines 66-68 was corrected "Mailhot et al. (2015) developed a relationship between annual duration of overflow occurrence and rainfall threshold category derived from rain gauge data, and described the value of rainfall data in terms of spatial co-location and temporal coincidence...."

2 Q. Line 98: "The focus of this study is the application..." Line 104: "... with (or by) weather radar...", Line 125: "phenomena" (if plural is what is intended) Answer. Line 98: One of the main purposes of this study illustrates application of local weather radar for CSO investigation in an urban watershed. The rain gauge data are used for the quality comparison or reference for the weather radar data. Therefore, the sentence of Line 98-99 is modified as "The focus of this study is application of weather radar in a small-scale sewershed allowing consideration of spatial variation." Line 104: The correction has been made. "Hydrometeorologic rainfall monitoring and measurement technology by weather radar has ..." Line 125: Yes. The intent is plural (multiple factors), wording modified to 'phenomena'.

3 Q. Line 141: Please specify more clearly what the correlation is here – do you mean the spatial correlation? Answer. Yes, implicit intention addresses spatial correlations. The temporal correlation temporal interval is defined as quarter-hourly (15 minutes) corresponding to the radar scan temporal resolution. The radar data available are radar volume scans (multiple rotations of the radar antenna at each scan angle) and this is transmitted each 4 to 6 minutes. The available shortest temporal resolution contains 3 to 4 instantaneous radar scans (including the base scan reflectivity). The results indicate a spatial correlation at 15-minute temporal resolution of near 0.6 at 5-km distance. The sentence was rephrased by "a spatial correlation near 0.6 at 5-km distance for ...."

4 Q. Line 144: What is the inter-gauge distance? Answer. Previously, the spatial structure of rainfall was evaluated using the rain gauge network over the same region by Hyun et al. (2016) as a case study. The local sewer agency, Metropolitan Sewer Dis-

trict (MSD), operates this rain gauge network. The inter-gauge distance represents the distance between neighboring gauges in the network. The overall average distance between neighboring gauges is about 11-km. The un-gauged site, indicated by rainfall (spatial) correlation less than 0.6, is 5-km away from the nearest gauge even in this relatively dense gauge network (15 well-maintained gauges are in operation for the study period of 2010-2014). Due to economic and practical reasons, many urban government entities such as sewer agencies face challenges in deploying and maintaining rain gauges in urban areas. The result is that the spatial value or useful limit boundary for single-gauge coverage is shorter than the necessary sub-hourly temporal resolution for this fine temporal resolution application. This leads the conclusion that appropriate use of fine-resolution radar rainfall estimates is necessary in such applications.

5 Q. Line 174: Could you justify this step a little further. If the central cell (of the 9) is smoothed to the average, is this not likely to remove what might be heavier rainfall? Alternatively this step could increase low rainfall values. Could you say how often this smoothing took place and how it may have affected the results? Could this be related to the underestimation you see, especially for tropical rainfall? Answer. The size of the 9-pixel radar region is about 100 acres. In the study by Hyun et al. (2016), correlation is above 0.9 within the 1-km distance range for quarter-hourly temporal resolution (attached figure shows the rainfall structure). This indicates the rainfall structure generally has limited variation within the 9 pixel region and smoothing influences are likewise expected to be minor. The main purpose of this data pre-processing step is to evaluate neighboring radar pixel values and account for discontinuous rainfall, if any, within the isotropic rainfall area. For all radar-rainfall data, the portion requiring adjustment (indicating more than 50% difference in neighboring pixels) to the raw value is only 0.068% across the range of 20-dBZ or higher reflectivity for the entire study period. In other words, the original data pixel value is available and used more than 99.9%. Based on this clarification, there is no expectation for underestimation in tropical type rainfall group.

6 Q. Line 178: Could you explain how the rainfall type is identified? Was this provided to you as part of the original dataset and the classification made by the providers of the data? Or is this what you mean by optimization. (As I have read further, I see on line 436 that this is discussed. Could you explain this earlier in the paper please?) Answer. Yes, the National Weather Service (NWS) selects the rainfall type from among these pre-defined types and assigns the type to a single radar site. The practical range of the radar is 150-km radius, and the entire coverage area is considered one rainfall type and this follows in conversion of reflectivity to rainfall using a single Z-R relationship. For this reason, this work is initiated with Level-II NEXRAD data from NWS rather than the higher level rainfall product which has been converted and smoothed using the pre-specified Z-R relation. Since any single storm may be dynamic with significant spatial variation, the pixel region corresponding to the catchment area or rain gauge site was individually identified by storm type for the 15-min temporal encapsulation period. The method of root-mean-square error (RMSE) with the rain gauge value was used to define the rainfall type each quarter-hour. The NWS or other agency does not act as a data provider to define rainfall type. In order to clarify this issue, a modification to the explanation of the decision making process for identification of rainfall type is added to line 178. "The KLVX radar data management system applies a Z-R relationship according to four storm types... with root-mean-square error (RMSE) every 15 minute individually."

7 Q. Line 189: Why is this equation in the appendix? Answer. All equations are now moved to appear within the text of the manuscript; and only notations remain in the appendix section.

8 Q. Line 205: Could you clarify what is meant by optimization here? Is it the selection of one of four Z-R relationships, or the modification of the parameters of the Z-R relationship? Answer. Basically, optimization in this work begins with the spatiotemporal downscaling for application of the rainfall type decision process. This is an important starting point for data quality evaluation and optimization. The tendency for rainfall un-

derestimation is observed in the original tropical rainfall group and is an undesirable result for hydrological applications. In order to reduce underestimation, the SVC process is a second or supplementary optimization and eventually leads to identification of two alternate tropical Z-R relationships. These two optimizations are complementary and both are necessary for the most optimal application of uncalibrated local radar data.

9 Q. Line 316: you talk quite early on in the paper about continuity, but it is not until line 432 where you define it. Could it be defined earlier? Answer. As you recognize, the paper describes two primary tasks – generation of radar rainfall and its application to investigation of combined sewer overflow. The first continuity concept addresses the identification of rainy periods into independent events; these then forms the series of "rain events" necessary for use in the CSO overflow evaluation. Later in the manuscript, use of the term "continuity" in Line 432, a more focused view considers each rain event individually. In this case, the uninterrupted occurrence in time or inter-connection of rainfall pattern for a single rainfall event is described as one factor influencing small-scale sewershed overflow occurrence. These two concepts are briefly described in the abstract and introduction sections. The authors are willing to make additional modifications to the text if recommended - Thank you for the comment.

10 Q. Line 319: Here you say super-resolution data is important to the estimation of CSO, but with your smoothing operation (already discussed), you may be losing the high resolution data that is important. If I understand correctly, the pixel resolution is 5 ha. If you use the smoothed result from 9 pixels, you are using the rainfall from an area of 45 ha. Is this correct? Answer. Yes, you are right. The radar is fundamentally use the polar coordinate system. The super-resolution radar resolution is 250-m and 0.5-degree (previously 1-km, 1-degree), and the radar site (KLVX) is 40-km from the study area. As in response no.6 to reviewer comment, rainfall variation is known to be limited across the 1-km range and each radar pixel is evaluated. However, there is a limited range effect along with distance from radar location. The radar range is

up to 230-km and at more distance locations the pixel size is influenced by smoothing and partial beam filling. The NEXRAD network is a two dimensional radar observation system and covering most of the entire the USA continent at a relatively fine radar resolution. As stated earlier, for this particular location, the use of any smoothing with surrounding pixels was limited in 0.068%. The NWS or National Oceanic Atmospheric Admistration (NOAA) provide conversion tools (NOAA Weather and Climate Tookit) to tranform the polar coordinate data into the Cartesian coordination system. The actual pixel size is 0.0020132 lat/lon degree (about 220m, the size of the pixel is about 12 acres). This implies that a single data value can be assigned to about a 200 acres area of the earth's surface at a quarter-hourly time interval. Although the spatial variation of rainfall depends on rainfall type, it was found to have limited to no influence for the study period.

11 Q. Line 370: I'm not sure I follow the analysis of acceptable and unacceptable over-flow events. Where there is an overflow and the overflow ratio is less than 0.6, is anything known about the volume of overflow that reaches the receiving body? What is an acceptable overflow? If I proceed to line 400, we read that for values greater than 0.4, there is a likelihood of significant pollution. Why then is 0.6 chosen as the threshold for acceptable overflow volumes? It is clear from Figure 11 that the unacceptable events are those events with greater volumes, but I would appreciate a better understanding of how you derive at this from the index. Answer. The coupling of the previously defined rainfall event and its corresponding overflow event is the most challengeable point in this study. The chosen CSO (CSO130) is isolated from neighboring sewersheds, and the flow data are only for the overflow discharge into the surface stream (Beargrass Creek, Louisville, KY). The flow data are provided by the local sewer agency and the flow records are known to contain overflow periods where there is no overflow – this is due to flow gauge error associated with debris or other disruption of the flow sensor. In order to select only overflow events with higher quality flow records, the flow data from the beginning of a rainfall event to the 6 hours after the end of rainfall were coupled as shown in Figure 7 (most significant nine overflow producing rainfall-overflow events).

Based on the EPA rainfall event definition, there must be no runoff producing rainfall for 6 hours following the end of a previous event. In this sewershed, the volume of dry-weather sewer flow alone is not influential for inducing an overflow and only the combination of sewer water and storm runoff together cause overflows. After identification of the rainfall-overflow events (coupling), we recognized existence of a limited number of excessive and non-rain event overflow records as shown in figure 8 (left). The partitioning or extraction of valid coupled rainfall-overflow data is necessary when selecting and using field data. In this case, a region in Figure 8 left-side, the range of ovwerflow ratio from about 0.6 to 0.9 shows 'no dots' (no events). This indicates that a threshold or runoff index may exist for this catchment with a value somewhere in this range (0.6 to 0.9). Additionally implying that higher runoff ratio values (above 0.9 and even exceeding 1.0) are likely due to flow gauge error sources. Therefore only coupled events which fall within the (0 to 0.6 (bolded-solid-quarter circle in figure 8) are included in the study. Following this index screening process, there remains 52 coupled rainfall-overflow events or "acceptable", and another 47 events were discarded as "non-acceptable".

Regulatory requirements from EPA prohibit municipalities from allowing CSO with frequency more than twice per year. This study formulates a systematic methodology to define the relation between rainfall spatiotemporal characteristics and CSO overflow. The methodology allows one to define a numerical rainfall volume associated with occurrence of overflow in a specific sewershed. Every sewershed would likely respond differently to the same type of rainfall depending on the size/land use/shape of the sewershed. This identification of overflow inducing rainfall characteristics in terms of volume, intensity, and temporal pattern for a specific sewershed allows development of strategies to mitigate overflow occurrence. For example, in this sewershed, the rain depth can be related to overflow as shown in Figure 9 (left). Additionally, discriminant analysis can divide overflow producing rainfall events into two groups – significant and non-significant. In figure 11, an index threshold line divides the 52 coupled events into two groups for discriminant analysis. The significant and noticeable group differences

are quantified by volume and duration of rainfall in Table 1. The traditional rainfall design methods, using IDF curves for example, are convenient but suffer from limitations when implemented to identify or characterize CSO occurrence. In short, this study allows a more specific characterization for the CSO overflow rain event, for example: "Occurrence of a 20-mm rainfall over a 6-hour period or a rainfall with intensity greater than 6-mm/15-min (24-mm/hr) will induce a significant combined sewer overflow event for the CSO-130 sewershed."

12 Q. Line 389: Could you say something about the relationship between rain gauge and radar for the CSO causing events? Are they in anyway different from the other rainfall events? Answer. The rain gauge data are from a gauge 700-m away from the sewershed, CSO 130. All gauge data was used as reference data for quality control, quality assurance and evaluation of the reflectivity-rainfall relationship. It is fortunate to have gauge data available close to the study site. In the period of 2010-2014, there are 15 quality-checked gauges is in operation across the city of the Louisville, Kentucky. According to Hyun et al. (2016), even though this is a relatively dense gauge network, applications in rainfall-runoff evaluation for a relatively small ungauged site still require radar-rainfall details due to the space-time variability of rainfall. For example, flash floods are recognized as an emerging disaster type due to local extreme storms in this climate changing era. In order to study the impact of the incremental changes in rainfall at the urban catchment scale, sub-hourly rainfall is essential. Therefore, the application and improvement of radar-rainfall estimation is an urgent need. The NOAA/NWS provide improved Level-III products such as dual-polarized instantaneous rainfall rate, but the product is not specified or optimized for local use. The future research plans must consider investigating use of operational dual-polarized rainfall products. Again, in this work, the rain gauge resource serves only as a reference in the evaluation of radar-rainfall; rainfall for the CSO event evaluation follows from the radar-rainfall estimation work.

13 Q. Conclusions: Could you discuss any differences you may have seen between the

two types of tropical storms? Can the information in your research be used to identify in real-time which Z-R relationship to use, and whether it can be used to improve the prediction of CSOs? The conclusions section could be improved. You say early on that "Categorization of the severe rainfall events including CSO occurrence can provide insights for hydrologic and hydraulic design guidelines to reduce sewer overflows from combined sewer systems in an urban area". Can you say a little more about how this might be done? Could you also say more about the number of false negatives in Table2? Out of 52 events listed, you predict 11 out of 52 incorrectly? What is different about these events. If you look at Table1, the major differences seem to be rainfall intensity and total depth (the other differences aren't great. This is what you would expect. When do you make a false prediction?

Answer. In the figure 5, the two newly created tropical Z-R relationships are plotted. Through use of the Support Vector Classification with Kernel threshold of the 46 dBZ, the two groups were divided by the linear hyperplane (lower-left in figure 4). Let's back to the figure 5 (lower-right), and the tropical 2 (red-dotted) power curve takes care the underestimated group. Therefore, the matching rainfall intensity to observed reflectivity is necessary to prevent the underestimation problem. Fundamentally, a single tropical type of Z-R relationship cannot cover storm variability in extreme convective. Addressing a shift in the fit of the Drop-Size-Distribution, in this case for the tropical storm due to the reflectivity threshold of 46 dBZ, resulted in an evenly distributed set of two groups.

Yes, clearly it is possible to apply these methods in near real-time weather prediction. However, processing speed is a key in practical radar operation. An efficient and well-designed multi-processor algorithm would make it possible to identify and assign rainfall type on a 'pixel-by-pixel' scale for the entire detection boundary and with a very short time window. However, the limits of current hardware and communications systems might hinder the operational usefulness. Nevertheless, details of storm dynamics are becoming more detectable by local weather observation and measurement networks and government agencies must continuously pursue the development of improved operations for understanding hydrologic processes in order to maintain public safety. Furthermore, a dense rain gauge will always be required to validate radar-rainfall for hydrologic applications. Recently more rain gauges (about doubled) has been deployed over the study area for the purpose of improving the "2 hour in advance weather forecasting" (MSD report 2016) for the city. In my opinion, only the densely populated areas need this "short-temporal rainfall-type assignment system" when considering the socio-economic effect. Improved accurate emergency weather forecasting may possible for a sub-section of the city or even smaller areas such as CSO sewersheds.

The conclusion section is modified as detailed below. These sections will be merged into the existing conclusions during the revision of the manuscript process.

In the Figure 7, the top 9 overflow events are plotted. Overall, rainfall volume governs the overflow, but the intensity (for example, CSO events 5, 6, and 7) and rainfall continuity (uninterrupted rainfall or event duration) (for example, CSO events 3, 4 and 8) also have a role. Rainfall volume, intensity, duration (uninterrupted rainfall) are historically important factors influencing hydrologic response. In a related study, a detailed study proposing a classification is being prepared for defining the classification of rainfall events under these guidelines. Results of that work indicate three different rainfall groups – high intensity group, high depth (volume) group and light rainfall groups. The groups were objectively partitioned through a k-means clustering method in the 2-dimensional intensity-volume rainfall field. The clustered groups are validated using an objective clustering performance measure. That work is currently in "review" status with the Journal of Hydrometeorology, American Meteorological Society [title: Rainfall event characterization with cluster and variogram analysis as an expedited contribution, Hyun et al.].

Discriminant analysis (DA) partitions the 52 coupled events into two groups according to the significance of the volumetric overflow. First, the discriminant analysis is used for

the categorical analysis, thus, the threshold of the 1.5 mm initially assigns the events evenly into one of two groups, significant or non-significant overflow. Based on the assigned factors: rainfall total volume, duration, rain peak, rain type, and rain continuity, DA predicts the events again. There are no false events in the DA analysis, but a relocation of group according to the assigned factors. Yes, each factor provides a part of the contribution to identify the discriminant among the group. The governing factor is volume, but this result does not mean volume is the only factor of significance in overflow occurrence. The EPA regulations on CSO overflow currently focus on a frequency based decision or rainfall event characteristic. However, this work hints that the volumetric approach may be more effective to identify the types of rainfall inducing a CSO. For these reasons, the DA algorithm is applied to distinguish significant overflows and identify the characteristics of the overflow inducing rainfall event.

The authors thank the referee for detailed comments which have provided guidance for an improved manuscript. The re-evaluation of the significance and presentation of the core findings has allowed this work to improve its conclusions in the context of weather radar optimization and applications to urban sewersheds. Thank you very much again.

Please also note the supplement to this comment:
http://www.hydrol-earth-syst-sci-discuss.net/hess-2016-362/hess-2016-362-AC1-supplement.pdf

———————————————

[Figure]

Fig. 1.

---

## Referee Comment (RC2) · Anonymous Referee #2 · 6 Oct 2016

The paper consists two parts, the authors first investigate different methodologies to translate reflectivity into intensity, which is not my expertise so I don't have specific comments regarding this part.

The second part is regarding the CSO analysis and there are several points unclear to me. 1. Could the authors define the "overflow depth"? Do the author mean the overflow water depth from CSO? or the runoff depth on the surface? If it is the casued overflow from CSO, it should be a volume divided by an area. How is the area defined? If it is the runoff depth on the surface, would "runoff depth" be more appropriate? However, the runoff depth is spatial varied and what is the location the authors are referring to? Also, the runoff depends on the terrain (slope, catchment area, etc.) and the drainage capacity, how did the author determine the depth? Also, in most cases, the overflow from CSO is unlikely to be linear relationship to rainfall depth when considering the

rainfall pattern, topography, catchment area and concentration time, drainage network and capacity. The ratio of "overflow depth" to "rainfall depth" is over-simplified and misleading.

2. The authors only discussed the relationship between the "rainfall depth" and "overflow depth" without considering the hydrology and hydraulics in the whole process. This is the major weakness of the paper. The authors said "a search to understand the contributing factors causing overflow events is warranted.", but the paper does not cover those critical contributing factors.

3. L352 The authors identified 95 rain events with coupled CSO occurrence in the sewershed. Is this a correct statement? Are those events selected according to EPA definition without considering CSO?

4. L370 Why it is not possible to determine the flow in sewer network and the overflow? Using sewer model can easily provide the answer. Otherwise, how did the authors get the overflow depth? If the rainfall-runoff index is a ratio of overflow depth to GAUGE rainfall depth, what's the point to estimate radar rainfall?

5. L441 How to determine the convective radar pixels and the total number of rain pixels? Are those only the pixels covering the sewershed? If yes, the sewershed is 13 ha and a radar pixel is about 5ha. So only up to 3 pixels are considered? If not, how will the rainfall outside the sewershed affect the flow in the sewershed?

---

## Author Comment (AC2) · 16 Oct 2016

Reply RC2(Anonymous Referee #2) on October 6th 2016

Authors: Jin-Young Hyun, Thomas D. Rockaway and Mark N. French

This document provided detailed response to referee comments (RC2) from Anonymous Referee #2. The authors recognize and thank this reviewer for the effort and suggestions to improve this manuscript.

1. Could the authors define the "overflow depth"? Do the author mean the overflow water depth from CSO? Or the runoff depth on the surface? If it is the caused overflow from CSO, it should be a volume divided by an area. How is the area defined? If it is the runoff depth on the surface, would "runoff depth" be more appropriate? However,

the runoff depth is spatial varied and what is the location the authors are referring to? Also, the runoff depends on the terrain (slope, catchment area, etc.) and the drainage capacity, how did the author determine the depth? Also, in most cases, the overflow from CSO is unlikely to be linear relationship to rainfall depth when considering the rainfall pattern, topography, catchment area and concentration time, drainage network and capacity. The ratio of "overflow depth" to "rainfall depth" is over-simplified and misleading.

Answer. Yes, each of these questions can be answered here. The "overflow depth" means the volume of water that is not captured by the sewer pipe and transmitted to the waste water treatment plant, and the volume is expressed as a depth (relative to the sewershed area size) as is commonly done in hydrologic applications. In general, for hydrologic studies, the water volume (rainfall or runoff or evaporation), and in this case the sewer overflow volume, can be defined or mathematically expressed as depth by dividing volume by watershed area. Yes, the overflow water depth is from the CSO outlet pipe measurement records from a gauge mounted at the discharge pipe. Yes, the overflow depth is expressed as a runoff depth using the definition noted earlier. During rainfall-runoff, stormwater runoff and sewage water mix together in the combined sewer system (CSS) and flow through the discharge pipe. As the sewer pipe begins to fill with this combined flow, the pipe capacity may be exceeded since in CSS pipes are typically not capable of conveying the rainfall-runoff and sewage flow. The result is an overflow event or combined sewer overflow. To prevent sewage backup into streets, residences, businesses, an overflow structure is used to divert excess capacity flow (volume) directly into surface waters. To accomplish this, the CSS pipe has a weir structure to divert water to the waste water treatment plant, and if flow depth exceeds the weir height, all overflow from the weir goes into a nearby surface water stream – and a Combined Sewer Overflow (CSO) event is recorded. As in many urban areas of the USA with CSS, these CSO events must be monitored – and now regulatory agencies such as the US Environmental Protection Agency (EPA) require municipal governments to eliminate all CSO events. The flow record for the overflow event includes measurement of the overflow volume since the overflow volume (not the portion transmitted to the waste water treatment plant) is the portion of interest since this runoff water (combined with sewer water) results in a public health issue and environmental pollution of surface waters. In this manuscript the term "overflow" is used in order to distinguish it from the portion of flow associated with stormwater only or sewage only. The CSO discharge is a mixture of these two water sources.

Yes, the drainage area is defined by topography of the land surface as the region draining or area contributing surface water runoff flow to the sewer pipe network draining to the most downstream pipe where the CSO diversion or overflow weir is located. Similarly, the drainage area also includes all "sanitary" sewer connections from area residential and business customers that contribute sanitary flow to the same pipe network draining to this common downstream CSO diversion point. If useful for clarification, the authors can add a figure or figures illustrating both the diversion weir and the specific catchment location. Surface runoff and sanitary sewage flows to this common downstream point and is the location where CSO measurement is recorded. The local sewer agency, Metropolitan Sewer District (MSD), Louisville, KY, identified the sewershed area after multiple surveys (The details uploaded in the supplement 'CSO 130: Combined Sewer Overflow Fact Sheet').

Yes, generally runoff depth is spatially varied in any catchment, however that is a not an aspect explicitly addressed in this work. Yes, runoff depth does depend on surface characteristics of the area as mentioned in this comment. However, the focus of this work is only on rainfall-runoff events that result in an overflow event – meaning the stormwater runoff combined with the sanitary sewer flow exceeds the CSS capacity. In cases where no CSO event occurs, all runoff (combined stormwater and sewage) flows to the waste water treatment plant and there is no public health or pollution issue. The runoff is measured at the sewershed drainage area outlet point (most downstream point) and since this area is entirely drained by a CSS, all runoff and sewer water flows past a common point in the pipe system. The CSO flow amount diverted to the

surface water stream is measured at the downstream (pipe) at the weir chamber. The volumetric overflow is then converted to depth for comparison in magnitude to rainfall depth for the storm causing the CSO event. It is common in hydrology to show both rainfall and runoff in a common unit such as volume or depth in order to visualize a relationship and evaluate quantities on a common scale. Again the volume of overflow water is converted to depth units by dividing by the drainage area.

The authors agree that in a typical rainfall-runoff study, one might attempt to identify a relationship to define a rainfall to runoff depth ratio in order to determine land-use runoff coefficients or develop a means to estimate surface runoff from rainfall. However that is not why the overflow depth was utilized here. Instead, the overflow depth was compared to rainfall depth as a direct means to evaluate the quality and significance of the CSO event. The goal of this study was to identify the type of rainfall events in terms of amount (volume, intensity, duration) associated with a CSO event. If one can identify such rainfall event characteristics, then municipal agencies can develop strategies to mitigate CSO occurrence. The EPA strongly prohibits the overflow of the combined sewer water into the aquatic environment, thus, it is valuable to investigate the overflow trend and pattern according to the extreme rainfall event. In most CSO sewersheds there is limited or no understanding of the specific rainfall patterns leading to CSO events in that particular location (downstream discharge point from the sewershed). Instead, agencies view the entire region and simple develop a relationship between regional rain depth and number of CSO occurrences across the areal region of responsibility such as a large urban area, a county, or other multiple watershed region. That type of view cannot provide optimized and most economical planning of mitigation approaches. In summary, as the reviewer states, earlier non-site-specific studies are oversimplified in some cases. That is the impetus for this work – here the authors propose and illustrate a non-simplistic, high-resolution, location-specific identification of rainfall characteristics that produce CSO events in a given sewershed. The framework developed and illustrated in this manuscript is universally applicable to CSS urban regions (give radar availability). With this approach, the results can be

used to assist with efforts to eliminate CSO events through the use of mitigation measures. Due to the variety of mitigation strategies available, and consideration of local hydrologic climate and regulations, specific measures are not proposed in this work – however the capacity of a mitigation strategy can be identified as outlined in the final summary table of results.

2. The authors only discussed the relationship between the "rainfall depth" and "overflow depth" without considering the hydrology and hydraulics in the whole process. This is the major weakness of the paper. The author said "a search to understand the contributing factors causing overflow events is warranted", but the paper does not cover those critical contributing factors.

Answer. The authors agree with this point, however, the direction of this work did not include development or identification of details of surface runoff characteristics or pipe flow hydraulics in the sewershed. The study applied a lumped-hydrology concept to identify rainfall event characteristics producing CSO events. This small-scale urban sewershed has high imperviousness. The study area is located in an urban residential and commercial area of the city of Louisville, KY, and according to MSD documentation the percent impervious is 71% with land-use estimated as [residential 24%, commercial 25%, industrial 32%, vacant 6%, undefined 13%]. Due to the relatively large percent imperviousness, the expectation is that rainfall volume governs overflow occurrence with limited influence from sewershed characteristics such as slope, antecedent condition, and other terrain features. As indicated in the runoff records, the short lag time (rainfall to runoff flow) and the high imperviousness, the influence of terrain is low for heavy rainfall (rainfall events identified according to EPA definition). In support of the short response time, the CSO event analysis was developed using sub-hourly temporal resolution to identify the 52 strongest storm events considered during the study period (Jan. 2011 to Dec. 2013). As stated in response to review question 1, this work proposes and illustrates a non-simplistic, high-resolution, location-specific identification of rainfall characteristics that produce CSO events in a given sewershed. The methods

are universally applicable to CSS urban regions (give radar availability), and results can be used to optimize CSO mitigation measures. Due to the variety of mitigation strategies available, and consideration of local hydrologic climate and regulations, specific measures are not proposed.

3. Line 352: The authors identified 95 rain events with coupled CSO occurrence in the sewershed. Is this a correct statement? Are those events selected according to EPA definition without considering CSO?

Answer. Yes, there are 95 overflow-producing rainfall events identified. Additionally, the events are screened as shown and explained in figure 8 to evaluate data quality. The EPA rainfall definition is specific for urban hydrology applications and yes, all these event are selected according to EPA but also encompass all overflow occurrences. The EPA rainfall events means the runoff-producing rainfall, 0.1 inch of rainfall for urbanized area – and is an independent evaluation of rainfall occurrence regardless of CSO occurrence. The specific consideration and selection of CSO occurrence is independent of the EPA rainfall definition.

4. Line 370: Why it is not possible to determine the flow in sewer network and the overflow? Using sewer model can easily provide the answer. Otherwise, how did the authors get the overflow depth? If the rainfall-runoff index is a ratio of overflow depth to GAUGE rainfall depth, what's the point to estimate radar rainfall?

Answer. The overflow was measured at the downstream point in the pipe at the diversion chamber overflow weir. This study did not address or evaluate the details of stormwater runoff in the sense of modeling or forecasting. The intent of this study, in brief, is to identify the specific characteristics of rainfall leading to a CSO occurrence in a specific sewershed. While the authors agree that a hydrologic model might be useful for other types of rainfall-runoff study, in this case it was not a necessary component since total runoff was not of interest. Since regulatory agencies such as the EPA, and the local agency responsible for sewage collection and treatment (MSD) have interest

only in the elimination of CSO events and that is the focus of this study. During rainfall-runoff events in which all water in the sewer flows to the waste water treatment plant (no CSO) – there is no need for consideration of the runoff amount. As stated in earlier responses above, the overflow amount is recorded at the downstream point in the sewershed discharge pipe. The specific location of the measurement is at the overflow weir where sewage flow combined with stormwater runoff is diverted into the surface stream. The EPA requires all sewer agencies in the US to identify and monitor all CSO locations, occurrences, and flow amounts. One impetus of this study is optimization of radar-rainfall for urban hydrologic applications. The reason for two rainfall sources in the screening of rainfall events is to minimize uncertainty. The multiple rainfall sources, both the locally-optimized radar and the nearest rain gauge were used. However, the ground-based rainfall measurement, rain gauge, was only used in evaluation of radar-rainfall estimation, and the filtering section. Rain gauge amounts were not used for the CSO runoff depth comparison – the rainfall depth for this part of the study was solely sourced from optimal radar-rainfall.

5. Line 441: How to determine the convective radar pixels and the total number of rain pixels? Are those only the pixels covering the sewershed? If yes, the sewershed is 13 ha and a radar pixel is about 5ha. So only up to 3 pixels area considered? If not, how will the rainfall outside the sewershed affect the flow in the sewershed?

Answer. The authors thank the reviewer for this comment. Yes, the spatial variation of rainfall was investigated and considered for this study. The results are summarize in the rainfall structure study by Hyun et al. (2016), where the spatial variation of the rainfall at quarter-hourly temporal resolution is found to be limited across this small-scale area. Additionally, the radar reflectivity data (radar pixels) and resulting rainfall values over the area do not vary relative to the immediate surrounding region. For this reason, as mentioned in the response to Anonymous Referee #1 (AC1: Reply on RC1 (comments from Referee #1) – question #10), the usefulness of the radar-rainfall spatial resolution is limited at this sewershed scale. However, for a

sewershed of greater extent (several square kilometers), the rainfall variation becomes apparent and must be incorporated. Since the authors did not know if the CSO associated rainfall spatial variability was significant, it was evaluated and considered. Again, due to the limited spatial extent of this sewershed the rainfall spatial variation is not an issue. The spatially-averaged radar reflectivity pixel data which falls across the study area (approximately three pixels) is used without areal weighting. This aligns with the evaluation of radar-rainfall over the sewershed using the nearest MSD rain gauge. The nearest rain gauge, TR05 which is 700 meters away from the study area, was used in the radar optimization component of this work. Implying that the radar data were optimized to provide ground level rainfall verified using this rain gauge. The radar reflectivity data were then transformed into rainfall using the optimal Z-R relationship and the spatial variation within this range was negligible. For this work the authors found limited variation for the rainfall events within the sub-kilometer range. Lastly, any rainfall outside the sewershed boundary will not directly influence the runoff in the sewershed of CSO130. Neighboring watersheds and sewersheds flow into other drainage networks whether it be natural streams, combined sewer systems (potentially with a CSO outlet), or into separate sanitary and storm sewer pipes. This sewershed is not connected on the surface and does not share any sewer pipe connection with adjacent catchments. The drainage region for this CSO is field-verified by the local sewer company.

Please also note the supplement to this comment:
http://www.hydrol-earth-syst-sci-discuss.net/hess-2016-362/hess-2016-362-AC2-supplement.pdf

**Supplement:**

**Combined Sewer Overflow Fact Sheet**

[Figure]

**MSD**
Metropolitan Sewer District

Louisville-Jefferson County
Metropolitan Sewer District

**CSO130**

**1400 STORY AVE**

[Figure]

**Record#:** 03211-1    **Budget ID:** H09141    **Budget:** $23,687,500

**MSD Atlas:** AW222    **Level of Control:\*** 0

**Map#:** MAL19    **Completion Date:** 12/31/2020

**Project#:** L_SO_MF_130_S_09B

**Project Name:** Story Avenue and Spring Street Green Infrastructure

**Station Name:** None

**Alt. Project#:** None

**Alt. Project Name:** None

**Alt. Project#:** None

**Alt. Project Name:** None

\* *Level of Control:*
  *Is the number of times that a CSO can discharge in a typical year to meet water quality goals,*
  *as determined by the Wet Weather Team Stakeholder Group.*

Map Data Source: LOJIC    \*Maps not to scale.

[Figure]

**CSO130**

[Figure]

Louisville-Jefferson County
Metropolitan Sewer District

Map Data Source: LOJIC     *Maps not to scale.

Report as of: 1/14/2014

WEBSTER ST

STORY AVE

| ● | CSO Location | ☐ | Buildings | ▲ | Pump Station | Combined Sewer Lines |
|---|---|---|---|---|---|---|
| – – | Overflow Lines | ▨ | Edge of Pavement | ○ | Manholes | SewerLines |
| | | | | | | Drainage Lines |

[Figure]

Louisville-Jefferson County
Metropolitan Sewer District

**CSO130**

Map Data Source: LOJIC      *Maps not to scale.

Report as of: 1/14/2014

[Figure]

ADAMS ST

CABEL ST

QUINCY ST

E WASHINGTON ST

FRANKLIN ST

E WASHINGTON ST

BOWLES AVE

WEBSTER ST

ADAMS ST

STORY AVE

FRANKFORT AVE

E WASHINGTON ST

64   64

BUCHANAN ST

BICKEL AVE

N SPRING ST

MELLWOOD AVE

E MAIN ST

| ● CSO In Service | ▢ CSO Boundary | → Combined | → Siphons | → Sanitary | → TP Effluent Lines |
| | | → Force Mains | → Overflows | → Trunk/Interceptors | → Air Release Lines |

[Figure]

**Fig 1:** The vehicles are parked in the parking lot between J. Gumbo's and the Medical Center Laundry

[Figure]

**Fig 2:** There are a lot of commercial vehicles around the job site.

[Figure]

**Fig 3:** The rungs have been corroded away although some still are present and could be harmful to entrants

[Figure]

**Fig 4:** The over flow line is in good condition

[Figure]

**Fig 5:** The downstream low flow line is in good condition

[Figure]

**Fig 6:** The upstream low flow line is in good condition. Most of the influent waste water is clear

[Figure]

**Fig 7:** The bar screen is showing signs of rust but no major affects to its structural integrity.

[Figure]

**Fig 8:** There is a space in the bar screen where debris could go through.

[Figure]

**Fig 9:** The bar screen is mounted securely over the dam.

[Figure]

**Fig 10:** The tide flex valve is shut. There is no S & F near the site

[Figure]

**CSO Characteristics Report**

[Figure]

**CSO Number:                    CSO 130**

| | |
|---|---|
| **CSO Name:** | Webster Street |
| **Overflow Type:** | Diversion Dam |
| **Solids and Floatables Device:** | Screens |
| **Drainage Area:** | 16 |
| **LOJIC Percent Impervious:** | 79% |
| **Modeled Percent Impervious:** | 71% |
| **Receiving Stream:** | Beargrass Creek |
| **General Location:** | South Fork Beargrass Creek |

| | |
|---|---|
| **Baseline AAOV (in MG/YR):** | 1.08 |
| **Number of Overflow Incidents (in Number/YR):** | 12 |
| **Average Duration of Overflow (in Hours):** | 2.33 |
| **Average Volume per Incident (in MGal/Incident):** | 0.09 |

| | |
|---|---|
| **LTCP 2014 AAOV (in MG/YR):** | 0.06 |
| **Number of Overflow Incidents (in Number/YR):** | 2 |
| **Average Duration of Overflow (in Hours):** | 1.333 |
| **Average Volume per Incident (in 1000 G/Incident):** | 0.03 |

| | |
|---|---|
| **Residential Landuse:** | 24% |
| **Commercial Landuse:** | 25% |
| **Industrial Landuse:** | 32% |
| **Parks:** | 0% |
| **Vacant Landuse:** | 6% |
| **Population Estimate:** | 118 |

**Results are based on models populated in December 2013 utilizing InfoWorks ICM Version 3.0. Models were populated with LOJIC data and supplemented with field survey data. CSO's listed having 0.0 Drainage Areas typically serve as system relief points. While a CSO may display 0.0 AAOV, note that this is based on typical year rainfall data.**

*Feb-14*

DATE: JUNE 4, 1991
CSO SERIAL NO. 130
REGULATOR NO. 31 - WEBSTER STREET SEWER
LOCATION: In parking lot south of Story Avenue opposite Webster Street
OVERFLOW TYPE: DAM
STRUCTURE OVERFLOW ELEVATION:
 426.70 - Crest of dam

OVERFLOW TYPE AND SEWER GEOMETRY:
The overflow structure is an abandoned regulator located on a 24" circular brick sewer flowing southeast on Webster Street.  A dam with a crest elevation of 426.70 diverts low flows through a 12" sewer at invert elevation 425.30 to the abandoned regulator chamber and exits through an 18" sewer to the 60" Middle Fork Trunk Relief Sewer.  Overflows top the dam and continue through the 24" brick sewer to a 27" concrete sewer to Beargrass Creek.  The invert elevation of the 12" sewer at the abandoned regulator chamber is 424.60.  The invert elevation of the 18" outlet sewer is 423.16.  The 18" sewer enters the Middle Fork Trunk Relief Sewer at invert elevation 419.35.  The invert elevation of the Middle Fork Trunk Relief at this point is 417.60.

BACKWATER LENGTHS AND STORAGE VOLUMES:
The backwater length is about 70 feet.  The storage volume is about 80 cubic feet.

DOWNSTREAM BACKWATER EFFECTS:
The overflow discharges to Beargrass Creek.  The elevation of Beargrass Creek at the discharge point is about 420.00.  High levels in the creek can backflow into the overflow pipe.
*NOTE: Regulator removed in 2007

Updated
Updated
12/08 (Revised Elevation per Survey)

Solids and Floatables Technology: Screen

REVISED ELEVATIONS PER JACOBI,
TOOMBS AND LANZ SURVEY, OCT. 2005
– JULY 2006 (NGVD 29)

[Figure]

RIM 445.90

MANHOLE

TOP OF SCREEN
EL. 427.80

TOP OF DAM
EL. 426.70

24" BRICK SEWER
I.E. 425.30

12" V.C.P.

REGULATOR
CHAMBER

18" V.C.P.

I.E. 424.60

I.E. 423.16

18" V.C.P.

12" V.C.P.

MAHNOLE

24" BRICK
SEWER

FLOW

175 FT± TO
STORY AVE

FLOW

SCREEN

DAM

FLOW

24" BRICK
SEWER

REGULATOR CHAMBER
(INTERNAL HARDWARE
REMOVED)

FLOW

27" CONCRETE
SEWER

TO BEARGRASS
CREEK

[Figure]

WASHINGTON ST

SITE
Serial No. 130

WEBSTER ST

QUINCY ST

STORY AVE ADAMS ST

BEARGRASS
CREEK

LOCATION MAP

**MSD**

Louisville and Jefferson County
Metropolitan Sewer District
700 W. Liberty Street
Louisville, Kentucky 40203-1913

502–587–0603  –  WWW.MSDLOUKY.ORG

**REGULATOR No. 31**

**WEBSTER STREET SEWER**

**OVERFLOW TYPE: DAM**

**SERIAL No. 130**

| | | Date | | | Scale |
|---|---|---|---|---|---|
| Made By | C.B.F. | 10–11–84 | Horiz. | | NONE |
| Checked By | T.M. | 1–16–90 | Vert. | | NONE |

---

## Referee Comment (RC3) · S. Thorndahl (Referee) · 20 Oct 2016

S. Thorndahl (Referee)

st@civil.aau.dk

The manuscript consists of two different parts. The first part is on adjusting Z-R relationships for radar rainfall estimation, and the second part analyses and identifies relations between rainfall characteristics and combined sewer overflow. Having conducted research with in both radar rainfall estimation and urban drainage, I initially thought the combination of the two interesting. Paper is well written and the approach is clear and understandable, there is however some major issues, explanations and assumptions that the authors in my opinion need to address, before the manuscript it is publishable in HESS.

1. Despite both subjects of the paper being very interesting, I don't see the point of joining them in one paper. Reading the paper there are in my opinion little that justifies

the use of radar rainfall. Same analysis of overflows could have been conducted with rain gauge data with similar or same results. I would suggest dividing the paper in two. One on the optimization of Z-R relationships and rainfall estimates, and one on the analysis of rainfall-overflow relationships. The objective of each paper would be much clear in this way.

2. You use a spatial resolution of 220 m of radar data, but a temporal resolution of 15 min. I would expect that a coarser spatial resolution is fairly sufficient, when you are using a temporal resolution of 15. min. or you should increase the temporal resolution in order to benefit from the fine spatial resolution. Since the radar data consists of instantaneous values every 15. min., an individual rain cell can move a large distance (and much more than the spatial resolution) within the time step of 15 minutes. Your estimation of the total precipitation over 15 minutes is thus probably not very accurate. See e.g. paper from this HESS special issue on "Rainfall and urban hydrology": Thorndahl et al. (2016) and references herein on the relationship between spatial and temporal resolution of radar rainfall data. In this paper there are also given references for advection interpolation (or downscaling) methods, which can convert the spatial resolution into temporal resolution, creating better volumetric rainfall estimates (e.g. Nielsen et al., 2014). Furthermore, I am missing the reason for looking at the adjacent 8 pixels. Why not just use one?

3. In the evaluation of Z-R relationships you discard rainfall intensities less than 5 mm/15 min. This is still a significant rainfall intensity, and I think this is problematic in terms of estimating lower rainfall intensities later on in the paper and especially since you have CSO spills generated with less rain than 5 mm. The large variability of the small intensities in figure 1 bottom right is probably related to the point above, that your intensity is not a sum over 15 minutes but a random instantaneous intensity within the 15 min window – and that the coincident rain gauge observation has a much better volumetric estimate of the rainfall over 15 minutes.

4. I am surprised that you don't consider traditional bias adjustment (e.g. Mean field

Bias) of radar rainfall data against rain gauges rather than adjusting Z-R relationship. There is substantial research conducted on this, applying different methods. I think you could get equally good estimates using a simple bias adjustment, without having to divide in different rainfall types, eg. as presented in Fig. 2.

5. Section 3: How is the overflow estimated? Is it measured? In that case I would be relevant to describe how and with some specifications of equipment. – Or is it modelled? Also it would be interesting to know how you define the overflow depth. I guess the overflow volume, per event divided by the contributing catchment area, e.g the impervious area. Is this the case? Please clarify. It could be relevant to discuss the timescale/timestep of the CSO estimates.

6. I am missing the point of using radar rainfall estimates to compare to the overflow depths. In fig. 6 it is evident that the 15 min. radar estimates have some spatial variability, but since the catchment you analyze is very small, I would not expect any significant spatial variability of rainfall within the catchment. In that case you could just use the rain gauge. – or do you think that the areal estimates are better provided by the radar?

7. I like the idea to try to characterize the rain producing overflow. I did similar analysis, based on modelling, e.g. trying to identify the impact of duration in CSO-volumes (Thorndahl, 2009). This might be relevant to compare to, even though the catchment characteristics and the upstream storage volume plays important roles. It could be relevant to mention the design criteria, if any, for CSO structures in terms of frequency of overflow, overflow volumes, etc?

References

Nielsen, J. E., Thorndahl, S. and Rasmussen, M. R.: A numerical method to generate high temporal resolution precipitation time series by combining weather radar measurements with a nowcast model, Atmospheric Research, 138, 1–12, doi:10.1016/j.atmosres.2013.10.015, 2014.

Thorndahl, S.: Stochastic long term modelling of a drainage system with estimation of return period uncertainty, Water Science and Technology, 59(12), 2331–2339, doi:10.2166/wst.2009.305, 2009.

Thorndahl, S., Einfalt, T., Willems, P., Nielsen, J. E., ten Veldhuis, M.-C., Arnbjerg-Nielsen, K., Rasmussen, M. R. and Molnar, P.: Weather radar rainfall data in urban hydrology, Hydrology and Earth System Sciences Discussions, 1–37, doi:10.5194/hess-2016-517, 2016.

---

## Author Comment (AC3) · 25 Oct 2016

Reply RC3(Referee: Dr. Søren Thorndahl) on October 20th 2016

Urban sewershed overflow analysis using super-resolution weather radar rainfall (Manuscript Number. hess-2016-362)

Authors: Jin-Young Hyun, Thomas D. Rockaway and Mark N. French

This document provided detailed response to referee comments (RC3) from Dr. Søren Thorndahl. The authors recognize and thank this reviewer for the effort and suggestions to improve this manuscript.

1. (Question) Despite both subjects of the paper being very interesting, I don't see the point of joining them in one paper. Reading the paper there are in my opinion

little that justifies the use of radar rainfall. Same analysis of overflows could have been conducted with rain gauge data with similar or same results. I would suggest dividing the paper in two. One on the optimization of Z-R relationships and rainfall estimates, and one on the analysis of rainfall-overflow relationships. The objective of each paper would be much clear in this way. (Answer) Yes, the manuscript contains two distinguished section; the initial submission was more limited in content regarding radar-rainfall and more on the CSO topic. In the earlier version, the authors provided a summary section for radar-rainfall estimation and it was more focused on the CSO and rainfall induced overflow analysis. Prior to the 'Interactive Discussion' period, the 'Editor Initial Decision' requested details to be included regarding the 'rainfall estimation' in order to provide explicit evidence of the innovative merit. Therefore, the significant sections on radar and rainfall and optimization supplement the first part of this paper. The authors agree to either continue this paper(s) in 'as-is' form or to develop two separate papers according to your suggestion. The authors accept either decision from the editor at the final stage of this discussion. The only concern is that if partitioned into two papers the review and publication schedule may be lengthen the publication period by another review process starting from the beginning.

2. (Question) You use a spatial resolution of 220 m of radar data, but a temporal resolution of 15 minutes. I would expect that a coarser spatial resolution fairly sufficient, when you are using a temporal resolution of 15 minutes or you should increase the temporal resolution in order to benefit from the fine spatial resolution. Since the radar data consists of instantaneous values every 15 minutes, an individual rain cell can move a large distance (and much more than the spatial resolution) within the time step of 15 minutes. Your estimation of the total precipitation over 15 minutes is thus probably not very accurate. See e.g. paper from this HESS special Issue on "Rainfall and urban hydrology": Thorndahl et al. (2016) and references herein on the relationship between spatial and temporal resolution of radar rainfall data. In this paper there are also given references for advection interpolation (or downscaling) methods, which can convert the spatial resolution into temporal resolution, creating better volumetric rainfall

estimates (e.g. Nielsen et al., 2014). Furthermore, I am missing the reason for looking at the adjacent 8 pixels. Why not just use one? (Answer) The Super-Resolution (0.5°, 250 m) National Weather Service (NWS) weather radar is used to retrieve the level II radar reflectivity (instantaneous base scan). The coordinate system was converted to Cartesian (220m by 220m) using Weather and Climate Toolkit which provided from National Oceanic Atmospheric Administration (NOAA). Yes, the authors absolutely agree to your statements. At the single radar pixel spatial scale, a temporal resolution less than 15-minute is better suited to limit storm advection variations. Fifteen minutes may allow advection across the distance of the radar pixel and the rain cell dynamics may change within the pixel boundary regardless of wind translation. This study does not consider a sub-pixel scale (pixel by pixel) approach to match rainfall values between the measured locations in the air and on the ground. In this microscopic view point, many other error sources may hinder synchronization such as falling time (even using base scan), and vertical wind variation. Current operational weather radar equipment at this location is sufficiently free of spatial limitations and constraints at the urban hydrologic sewershed addressed by this case study. A study of the spatiotemporal variation of the rainfall, Hyun et al. (2016) provides objective measures evaluated to determine the validity of the spatiotemporal extent of the resolutions utilized. For this specific location, for the selected study period, at 15-minute temporal resolution, a high correlation is observed across a few kilometers of range. The authors used ground-based gauge data to evaluate this relationship, and the nearest-neighbor inter-gauge distance is 4.9 km. Even though this assumption is based on an extrapolation of correlation structure (attached figure in AC1 in the discussion), it supports with confidence the conclusion that rainfall for these events tends toward an isotropic condition across several adjacent pixels in any direction. The study area sewershed is 29 acres (11 hectares), and thus the isotropic condition is expected based on the spatiotemporal structure of rainfall for these events. In Thorndahl et al. (2016), temporal and spatial resolution of rainfall data were utilized for types of hydrologic applications somewhat different from this work. This study was not as concerned with the broad topic of urban rainfall observation network optimization, but only with the optimization of available data records to identify rainfall characteristics at a specific site. Meaning, if a rain gauge(s) were located directly in the sewershed it could have served as a primary source for volume estimation, and radar-derived rainfall taken less of prominent role. To meet this requested modification, the authors suggest the sentence may be rephrased as '...a 15-minute resolution used for the overflow analysis portion of this work since this is an event-based approach and regional rainfall structure details indicate a degree of homogeneity within the spatiotemporal rainfall structure (Hyun et al. 2016)". In summary, the authors thank the reviewer for this comment and note the recommendation of considering a 10-minute temporal resolution to minimize influences of rainfall advection. The reviewers research (Thorndahl et al. 2016) addresses a spectrum of fascinating topics of urban hydrologic research and the authors would like to add citation of this reference to note the considerations of these effects in the urban application of the radar-rainfall products. In summary for this study, only 0.068 % of the observation periods required utilization of a neighboring radar data when the target pixels covering the sewershed were not available. In other words, most of the data were derived from the values of the radar reflectivity directly over the sewershed.

3. (Question) In the evaluation of Z-R relationships you discard rainfall intensities less than 5mm/15min. This is still a significant rainfall intensity, and I think this is problematic in terms of estimating lower rainfall intensities later on the paper and especially since you have CSO spills generated with less rain than 5mm. The large variability of the small intensities in figure 1 bottom right is probably related to the point above, that your intensity is not a sum over 15 minutes but a random instantaneous intensity within the 15 minute window – and that the coincident rain gauge observation has a much better volumetric estimate of the rainfall over 15 minutes. (Answer) Again, the authors thank you for your comments on this topic. Yes, a 5 mm depth can be a significant amount of the rainfall. The figure 1 does not discard any rainfall observations. The figure 1 (lower-right) is a scatter plot using NWS Z-R conversion (least RMSE error), and all the retrieved scan reflectivity. The discarding of rainfall values less than 5 mm over

a 15 minute interval was performed only for calibration or identification of the optimal Z-R relationship in order focus on extreme rainfall events and in order to minimize underestimation issues at this temporal scale. Yes. The rain gauge network is operated by the local sewer agency, Metropolitan Sewer District (MSD), and the nearest gauge is deployed about 700 m away from the study region. Based on the spatiotemporal correlation structure identified by Hyun et al. (2016), the gauge is suitable for use as a volumetric data source. However, also fortunate in this case is that this gauge is located within a kilometer range since the average intra-gauge-network distance is near 11 km, implying that a single gauge data provides rainfall coverage for more than a 5km range. At a 15 minute of temporal scale, the spatiotemporal characterization definition by a single rain gauge is expected to be low. Meaning, at the fine temporal resolution required for most urban hydrologic runoff related studies, there remains many sewersheds not covered by ground-based rainfall (gauge) locations, even though gauge networks are considered densely deployed. Broadly, this study countered the negative effect of gauge location and temporal downscaling through application of locally calibrated radar-rainfall products at a scale appropriate for the urban watershed.

4. (Question) I am surprised that you don't consider traditional bias adjustment (e.g. Mean field bias) of radar rainfall data against rain gauges rather than adjusting Z-R relationship. There is substantial research conducted on this, applying different methods. I think you could get equally good estimates using a simple bias adjustment, without having to divide in different rainfall types, e.g. as presented in Figure 2. (Answer) Yes. The use of traditional methods was considered during this research. One intriguing aspect of this study was optimization of radar-rainfall for the localized urban area (sewershed). The mean-field-bias is a typically a single correction (up or down) and may not account for rain gauge locations in the vicinity of a specific site, such as this single sewershed, since adjustment is uniform across the region. There are more advanced and detailed MFB methods (Thorndahl et al. 2016) but again, additional complexity of analysis is required. This study acknowledges the existence of rainfall spatial variation across the areal extent associated with the meso-scale city region, and this is why the

authors adopted a site specific type of radar optimization process rather than a single overall radar calibration. The NWS/NOAA incorporates four fixed empirical Z-R relationship across the entire US nation. Furthermore these traditional relationships do not necessarily recognize more recent types of rainfall patterns and possible influences of climate change on rain cloud characteristics. This study focus on the suitability of the Z-R relationship into the localized area at a fine temporal resolution. As results indicate, it is beyond the capacity of a few operational Z-R relationships to adequately represent extreme storms (tropical storm region). For this reason, an investigation was completed to develop additional relationships for extreme storm types. As shown in figure 4 (tropical storm scatter plots), a single correction for the otherwise underestimated rainfall produces an overestimation problem. For this reason, a partitioned approach, as a sort of mean micro-field bias (MFB) adjustment, was developed specifically for the projects purpose. Since the data resources and capability exists to show this concept and illustrate its application, this is the reason to overstep the traditional single MFB method. The authors recognize that for larger regions, such as when the entire radar site (150-230km radius) or large watershed are considered, it may be more suitable to use traditional calibration methods such as MFB. Thank you very much again for this discussion point.

5. (Question) Section 3: How is the overflow estimated? Is it measured? In that case I would be relevant to describe how and with some specifications of equipment. – Or is it modelled? Also it would be interesting to know how you defile the overflow depth. I guess the overflow volume, per event divided by the contributing catchment area, e.g. the impervious area. Is this the case? Please clarify. It could be relevant to discuss the time-scale/time-step of the CSO estimates. (Answer) Yes. The overflow was measured at the downstream point in the pipe at the diversion chamber overflow weir (and the amount includes only the measured overflow amount directed into the surface stream – this is due to the requirement by the Federal Government agency, EPA (Environmental Protection Agency) to have a report on the overflow volume amount). The overflow water volume is measured from a gauge mounted at the discharge pipe.

These data records are obtained from the local sewer agency (MSD mentioned earlier) and measurement devices were not deployed specifically for this study. As with the rain gauge network and the weather radar records, the specific type of gauges, manufacturer, data logger and data transmission specifications are not readily available to the authors, but were requested from the MSD agency. It is expected that the equipment is representative of standard types used around the USA (or globally) for sewer pipe flow. Typically these are weir-type gauges with a pressure sensor to detect water depth, which is then converted to flowrate. The flowrate integrated over time provides the overflow volume. The "overflow depth in mm" means the volume of water that is not captured by the sewer pipe and transmitted to the waste water treatment plant, and the volume is expressed as a depth (relative to the sewershed area size – dividing volume by the sewershed size). The coupled data of rainfall and overflow were integrated into a single "coupled event" through the terminology "rainfall event" as defined by US Environmental Protection Agency (USEPA) and the hydrologic lag-time is not a directly significant in this study due to the small size of the sewershed.

6. (Question) I am missing the point of using radar rainfall estimates to compare to the overflow depths. In figure 6, it is evident that the 15 minute radar estimates have some spatial variability, but since the catchment you analyze is very small. I would not expect any significant spatial variability of rainfall within the catchment. In that case you could just use the rain gauge. – or do you think that the areal estimates are better provided by the radar? (Answer) Thank you for your comment. The authors will address the point more clearly and make a modification in the manuscript statements to do so. The study area of catchment CSO130 is small, but also isolated from all other sewer networks. Therefore, it is possible to investigate the relationship of rainfall and overflow directly. This also allowed consideration of this small sewershed region as isolated from surroundings instead of a typical case with a larger catchment where rainfall variability might be a factor (or sewersheds connected to one another and therefore a need to consider upstream inflow amounts). Therefore, it also allowed illustration of the benefit of radar-rainfall in the urban setting by describing some of the limitations of one-dimensional rainfall measurement equipment, rain gauges. Of course, since the rain gauge records rainfall volume near the surface, in many instances, when co-located, it is advantageous to use rain gauge records. However, as the distance between the gauge location and the catchment area of interest increases, consideration of the critical spatial variation distance, in conjunction with the temporal scale of interest must be weighed. In this work the temporal scale is 15 minutes, and the nearest gauge distance for a research study must be not more than a kilometer. It means that the urban setting needs an extremely dense network of the rain gauges in order to provide high quality rainfall definition. Realistically, these conditions are unlikely due to several issues including obtaining permission for access to deployment locations and maintenance issues. Fortunately, this catchment has well-maintained rain gauge 700 m from the study area, however, even due to this distance, this gauge data were only used for referencing the radar data calibration. Additionally, the second closest rain gauge, TR12 (lower-left plot in figure 6) was compared. Based on that, the two-dimensional radar-rainfall estimation was indicated as more suitable for this downscaled/microscopic urban hydrologic study.

7. (Question) I like the idea to try to characterize the rain producing overflow. I did similar analysis, based on modelling, e.g. trying to identify the impact of duration in CSO-volumes (Thorndahl, 2009). This might be relevant to compare to, even though the catchment characteristics and the upstream storage volume plays important roles. It could be relevant to mention the design criteria, if any, for CSO structures in terms of frequency of overflow, overflow volumes, etc? (Answer) The authors really appreciate for your encouragement. Currently, US EPA regulate the prohibition of CSO events by indicating simply a frequency (return period) basis of the rainfall events causing overflows. However, this regulation is too simplified an approach and does not consider magnitude of the overflow volume. Due to the spatial variation of rainfall, duration of rainfall, and size of the surface stream impacted by the overflow volume, a single overflow event may cause serious degradation of the aquatic environment due to a high concentration of harmful constituents in the combined

sewer water. The volumetric approach is necessary for environmental hydrologic mitigation of overflows as well as for design/modeling. This study provides a more specific characterization for the CSO overflow rain event, for example: "Occurrence of a 20-mm rainfall over a 6-hour period or a rainfall with intensity greater than 6-mm/15-min (24-mm/hr) will induce a significant combined sewer overflow event for the CSO-130 sewershed." This wording could be included in the revised manuscript conclusions. The authors appreciate the reviewer for these comments and information.

Please also note the supplement to this comment:
http://www.hydrol-earth-syst-sci-discuss.net/hess-2016-362/hess-2016-362-AC3-supplement.pdf